# Heterogeneous somatostatin-expressing neuron population in mouse ventral tegmental area

Elina Nagaeva[1], Ivan Zubarev[2], Carolina Bengtsson Gonzales[3], Mikko Forss[1], Kasra Nikouei[3], Elena de Miguel[1], Lauri Elsilä[1], Anni-Maija Linden[1], Jens Hjerling-Leffler[3], George J Augustine[4], Esa R Korpi[1]*

[1]Department of Pharmacology, Faculty of Medicine, University of Helsinki, Helsinki, Finland; [2]Department of Neuroscience and Biomedical Engineering, Aalto University, Espoo, Finland; [3]Department of Medical Biochemistry and Biophysics, Karolinska Institutet, Stockholm, Sweden; [4]Lee Kong Chian School of Medicine, Nanyang Technological University, Singapore, Singapore

**Abstract** The cellular architecture of the ventral tegmental area (VTA), the main hub of the brain reward system, remains only partially characterized. To extend the characterization to inhibitory neurons, we have identified three distinct subtypes of somatostatin (Sst)-expressing neurons in the mouse VTA. These neurons differ in their electrophysiological and morphological properties, anatomical localization, as well as mRNA expression profiles. Importantly, similar to cortical Sst-containing interneurons, most VTA Sst neurons express GABAergic inhibitory markers, but some of them also express glutamatergic excitatory markers and a subpopulation even express dopaminergic markers. Furthermore, only some of the proposed marker genes for cortical Sst neurons were expressed in the VTA Sst neurons. Physiologically, one of the VTA Sst neuron subtypes locally inhibited neighboring dopamine neurons. Overall, our results demonstrate the remarkable complexity and heterogeneity of VTA Sst neurons and suggest that these cells are multifunctional players in the midbrain reward circuitry.

*For correspondence:
esa.korpi@helsinki.fi

**Competing interests:** The authors declare that no competing interests exist.

## Introduction

The ventral tegmental area (VTA) in the mammalian brain is the origin of dopaminergic projections in the mesolimbic and mesocortical pathways (*Yamamoto and Vernier, 2011*). Proper function of these pathways is required for important behaviors such as motivation and learning (*Schultz et al., 1997*; *Wise, 2005*), with disturbances being associated with several pathological states including addiction among others (*Korpi et al., 2015*). Therefore, substantial effort has been invested in characterizing VTA dopamine (DA) neurons, so we now understand the heterogeneity, connectivity and function of these cells (*Björklund and Dunnett, 2007*; *Watabe-Uchida et al., 2012*; *Beier et al., 2015*). However, the VTA also harbors a sizable population of neurons expressing γ-aminobutyric acid (GABA) and a smaller population of glutamatergic (Glu) neurons (*Morales and Margolis, 2017*). Indeed, VTA GABA neurons are known to regulate prediction error detection by VTA DA neurons (*Eshel et al., 2015*). Further, optogenetic photostimulation of VTA GABA neurons (identified by expression of glutamic acid decarboxylase 65 or the vesicular GABA transporter, *Slc32a1*) strongly inhibits local DA neurons via GABA$_A$ receptors, resulting in aversive place conditioning and reduced reward consumption (*Tan et al., 2012*; *van Zessen et al., 2012*). Activation of VTA GABA projection neurons that inhibit cholinergic interneurons in the nucleus accumbens (NAc) also increases discrimination of aversive stimuli (*Brown et al., 2012*). Finally, inhibition of VTA GABA neurons appears to

be an important mechanism of action for pharmacological agents, such as opioids and GABAergic compounds (*Johnson and North, 1992*; *Tan et al., 2010*; *Vashchinkina et al., 2014*).

In the cortex and other forebrain regions, there is ample evidence for great heterogeneity in interneuron subtypes (*Rudy et al., 2011*; *Pelkey et al., 2017*; *Krabbe et al., 2018*; *Tasic et al., 2018*). To date, research on VTA GABA interneurons has focused on their functions without determining whether distinct subtypes exist. In the mouse cortex, the most common interneuron subtypes include somatostatin (Sst), parvalbumin (PV) and 5HT$_{3A}$ receptor-expressing neuronal populations (*Rudy et al., 2011*), which can be manipulated using mouse Cre-lines (*Taniguchi et al., 2011*). This approach has allowed a deeper understanding of the cortical interneuron subtypes, by establishing their intrinsic electrical properties and gene expression profiles, as well as their physiological roles in circuit functions (*Yavorska and Wehr, 2016*). Indeed, even GABA neuron populations in the forebrain are not identical in all regions; for example, GABA neuron subtypes in the basolateral amygdala differ from those of the cortex (*Krabbe et al., 2018*). Thus, translating the properties of GABA-neuron subtypes across brain regions is challenging, especially for brain regions as evolutionarily distant as the midbrain and the cortex (*Achim et al., 2012*).

Here, we focused on the Sst subtype of VTA neurons, which are fivefold more abundant than PV neurons and 50-fold more abundant than neurons expressing vasoactive intestinal peptide (VIP) (*Kim et al., 2017*). Moreover, in the forebrain, Sst neurons represent the most molecularly-diverse GABA neuronal population [more than 20 Sst-subtypes in the mouse visual cortex (*Tasic et al., 2018*)]. By using a Sst-Cre mouse line and combining electrophysiological assays with morphological reconstruction, single-cell sequencing (PatchSeq) and optogenetic circuit mapping, we have distinguished three distinct subtypes of VTA Sst neurons. We also found that some VTA Sst neurons express not only markers of GABA neurons, but also express markers for Glu and/or DA neurons. These results reveal that VTA Sst neurons are remarkably heterogeneous.

## Results

### Numerous somatostatin-expressing cells in the VTA

We estimated the number of Sst-expressing cells by using both immunohistochemical and in situ hybridization approaches (*Figure 1*). Anatomically, Sst neurons formed two distinct clusters within the VTA area: lateral and medial. The lateral cluster was located in the largest VTA nucleus, the parabrachial pigmented (PBP) nucleus, while the medial cluster was in the more caudal part of the VTA, within the parainterfascicular (PIF) and paranigral nuclei (PN) (*Figure 1a*). Initial cell counting using tyrosine hydroxylase (Th) immunohistochemical staining and Sst-Cre-linked cell marker expression (*Figure 1—figure supplement 1*) yielded an estimate for the number of VTA Sst cells of approximately 12% of the number of Th-positive neurons (809 ± 205 cells vs. 6800 ± 1080 cells, n = 3, here and after mean ± SEM, from bregma −2.8 to −3.8 mm). About 10% of the Sst-positive cells co-stained with a Th antibody. The density of Sst cells increased in the rostrocaudal direction, starting from bregma −3.5, which was the opposite to that of Th+ neurons, which decreased in number starting from the same level (*Figure 1b–c*).

We extended our analysis of Sst cells via the robust RNAscope in situ mRNA hybridization method (*Wang et al., 2012*). Examination of six coronal sections (12 μm thickness) at different bregma levels, confirmed a rostro-caudal gradient in Sst neuron density (*Figure 1c*). Estimates of the total number of cells were higher using this method (*Sst*: 1120 ± 470 cells vs. *Slc6a3*: 14600 ± 1610 cells, n = 4, from bregma −3.07 to −3.88 mm) and is in line with recent estimates of 1500 Sst neurons in the VTA of adult mice (*Kim et al., 2017*).

### Three electrophysiologically distinct subtypes of Sst neurons

Measurements of the intrinsic electrical properties of Sst neurons revealed the presence of three distinct subtypes of Sst cells in the VTA of juvenile P17-P23 mice. The names we have assigned these neurons are based on their distinctive features: *afterdepolarizing* (ADP) neurons demonstrated a prominent afterdepolarization (13.8 ± 0.5 mV, n = 215) after an action potential (AP; *Figure 2a*); *high-frequency firing* (HFF) neurons had the highest maximal rate of AP firing (129 ± 5 Hz, n = 92; *Figure 2b*); while *delayed* neurons fired APs with a significant time delay (279 ± 20 ms, n = 85) after the start of a depolarizing current, even at the high levels of depolarization (*Figure 2c*) (*Video 1*).

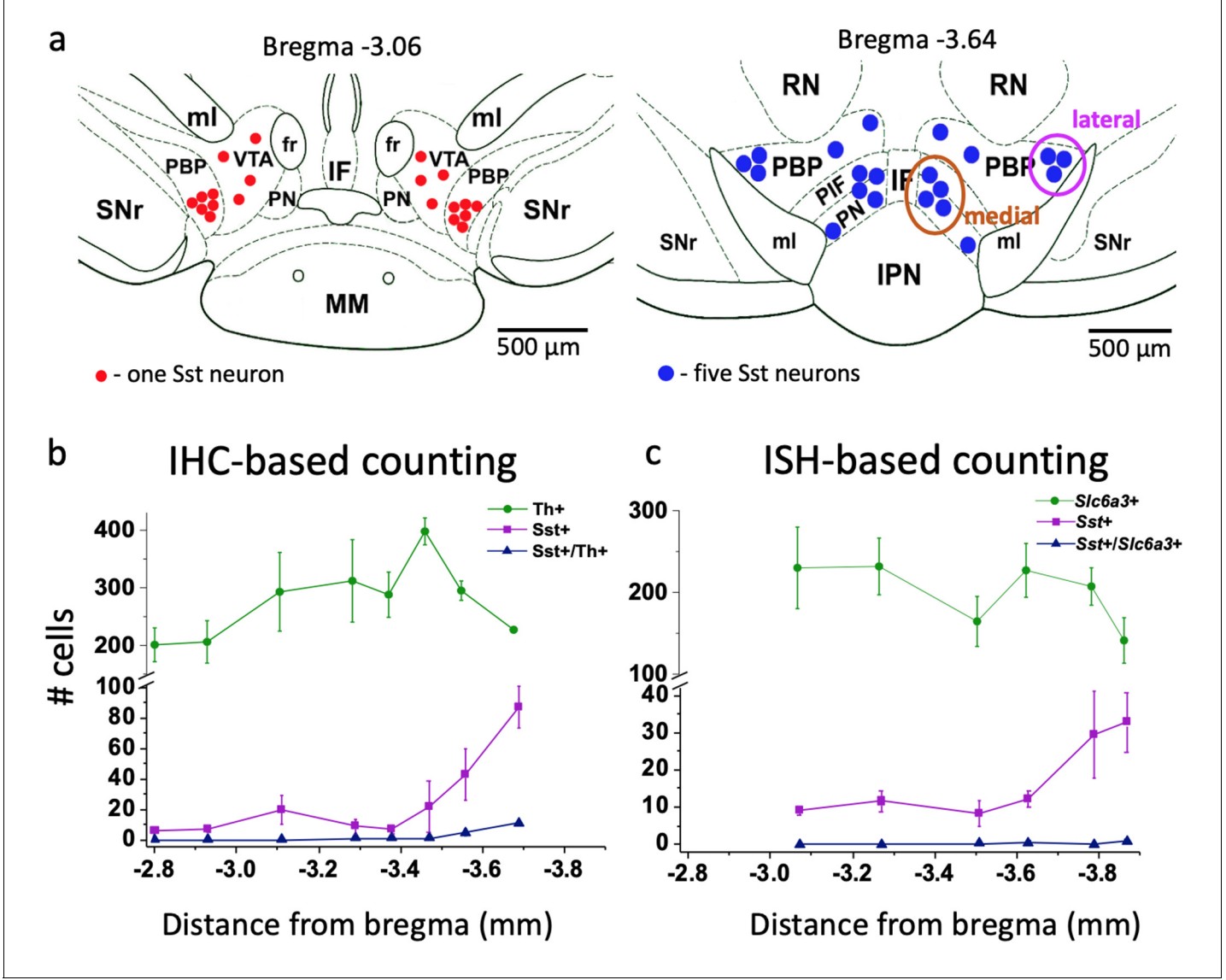

**Figure 1.** Localization and number of Sst and DA neurons in the mouse VTA. (a) Schematic representation of lateral and medial Sst neuron clusters in coronal sections at different distances from bregma: the left figure represents the rostral VTA with only lateral clusters and small number of cells (each red dot = one Sst neuron); the right figure of the caudal VTA shows medial (PIF+PN) and lateral (PBP) cell clusters and a higher number of cells (each blue dot = 5 Sst cells). IF, interfascicular nucleus; IPN, interpeduncular nucleus; MM, medial mamillary nucleus; PBP, parabrachial pigmented nucleus; PIF, parainterfascicular nucleus; PN, paranigral nucleus; RN, red nucleus; SNr, substantia pars reticulata; VTA, ventral tegmental area (rostral part); ml, medial lemniscus; fr, fasciculus retroflexus. (b) Cell counts for DA and Sst cells using immunohistochemical (IHC) approach: Th-antibody staining for DA neurons combined with inbuilt dTomato signal for Sst neurons of Sst-tdTomato mice. Number of cells is given per 40-μm-thick coronal section as mean ± SEM (n = 3 mice) at different levels from bregma along the rostro-caudal axis. (c) Cell counts, using in situ hybridization (ISH) approach with RNAscope probes for *Sst* and *Slc6a3* mRNAs. Number of cells is given per 12-μm-thick coronal section as mean ± SEM (n = 4 mice). The number of Sst neurons (magenta) increased in more caudal part of the VTA with both staining methods. Supporting data can be found in the Additional files: *Figure 1—source data 1*. *Figure 1—figure supplement 1*.

The online version of this article includes the following source data and figure supplement(s) for figure 1:

**Source data 1.** Raw data for *Figure 1b–c*.

**Figure supplement 1.** Anatomical localization of Sst neurons within the VTA in coronal plane.

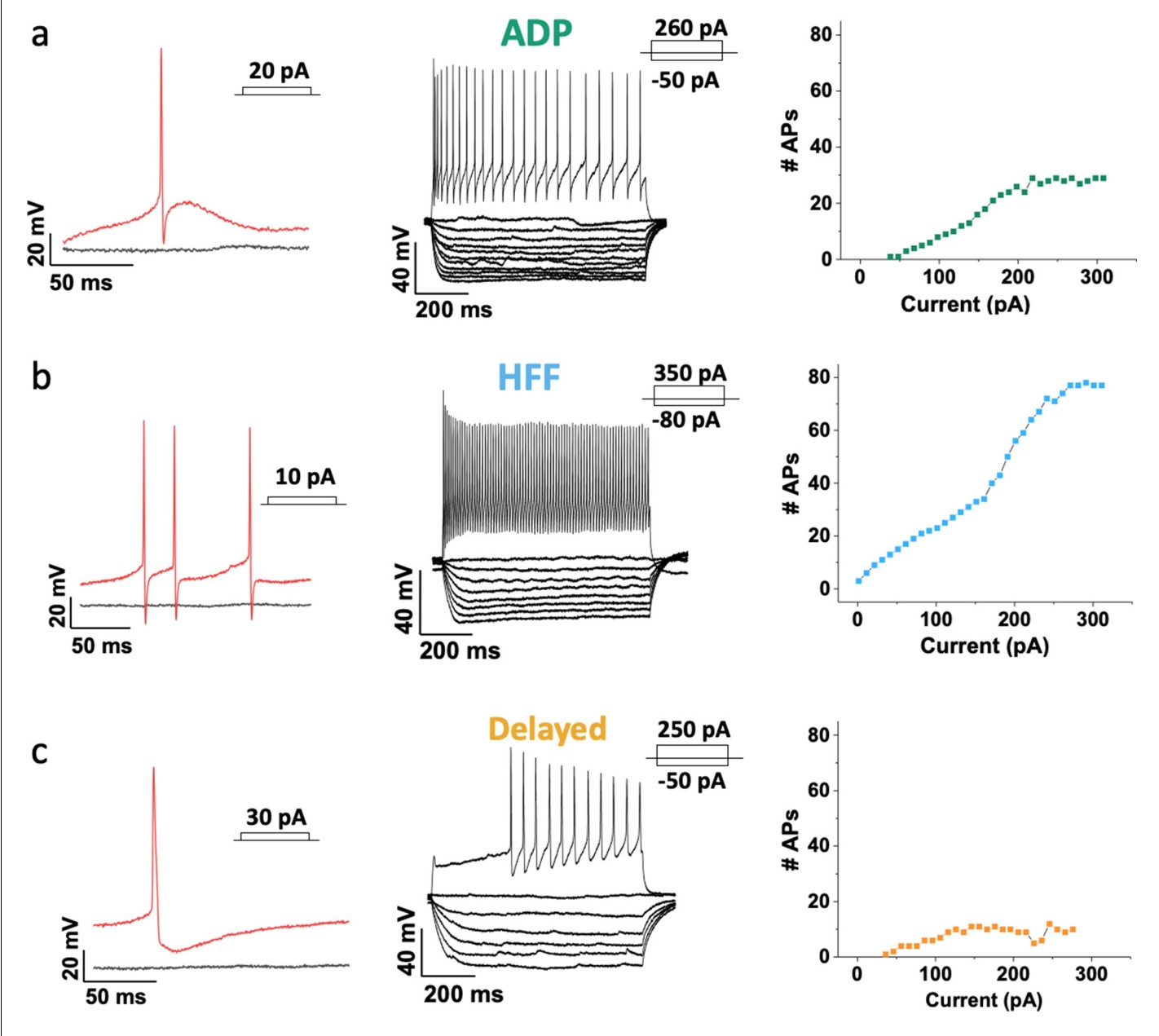

**Figure 2.** Three major Sst-expressing neuron subtypes in the VTA based on their firing properties. From left to right: example traces of the action potentials (APs) (in red) evoked by rheobase current injection, black line represents the baseline with no current injected; example traces of subthreshold voltage-responses of the same cells to 800 ms current steps with 10 pA increment, together with voltage-responses to saturated excitation; cumulative spike counts after increasing currents steps. (a) ADP neurons showed salient afterdepolarization (ADP) at rheobase current and a visible adaptation in firing at the saturated level of excitation. (b) High-frequency firing neurons (HFF) displayed sharp prominent afterhyperpolarization at rheobase current and the highest number of APs at the saturated level of excitation. (c) Delayed neurons demonstrated a long delay preceding the firing at the saturated level of excitation. Slow afterhyperpolarization and low adaptation were distinctive features of this particular subtype.

The presence of three neuron subtypes was confirmed by combining principal component analysis (PCA) of 25 electrophysiological features with unsupervised clustering via a Gaussian mixture model (GMM). The resulting model was tested with a Bayesian information criterion (BIC), which identifies the best combination of PCs and number of clusters (total n = 392; *Figure 3a–b*, *Figure 3—figure supplement 1*). Our clustering algorithm also revealed additional electrical properties that distinguish these subtypes of Sst neurons (*Figure 3b*, *Table 1*, *Figure 3—figure supplements 2–4*). Thus, HFF neurons had the lowest current thresholds and produced the largest number of APs at both the lowest and highest levels of excitation (3.9 ± 0.2, $F_{(2, 389)}$=80.71; p<0.0001, and 39.1 ± 3.1, $F_{(2, 389)}$=58.70; p<0.0001, during 800 ms depolarization) as compared to other two subtypes. Brief, prominent afterhyperpolarizations (AHPs) and the narrowest APs were also distinctive features of this neuron subtype, consistent with the ability of these cells to fire APs at high frequency. Delayed neurons showed low excitability, with the highest threshold current (40.8 ± 4.3 pA, $F_{(2, 389)}$=46.83; p<0.0001), broadest and slowest decaying APs. In addition to their prominent ADPs, ADP neurons tended to have lower input resistance (783 ± 24 MΩ, $F_{(2, 389)}$=5.121; p=0.0064) and higher level of AP frequency adaptation (0.28 ± 0.02); otherwise, these neurons were intermediate in their electrical properties and were the most abundant Sst neuron subtype.

To examine whether these three electrophysiological subtypes are restricted to only early developmental stages, we also recorded from an additional 92 Sst neurons from adult (P90) animals and added them to the dataset of juvenile mice. Re-clustering of the data defined the same three clusters and the 92 additional neurons (black circled dots in *Figure 3a*) mingled with juvenile cells, indicating that the subtypes remained the same in young (P20) and adult (P90) animals.

## Anatomical distinctions between three neuron subtypes

Most ADP (64%) and HFF (71%) neurons were found in the lateral VTA, predominantly in the PBP nucleus (*Figure 4a–b*), whereas Delayed neurons were primarily in the medial part of the VTA (79%) and almost equally distributed between the PIF and PN nuclei. From a ventrodorsal view, a larger proportion of ADP (71%) and HFF (59%) neurons resided within the ventral VTA, while most Delayed neurons (84%) were located in its dorsal part. This segregation of locations suggests that the different Sst neuron subtypes may subserve different functions.

## Morphology of VTA Sst neurons

The morphology of VTA Sst neurons was determined by loading 49 cells with biocytin or neurobiotin, followed by 3D reconstruction of their structure. The cells had a relatively simple structure, with thin and faintly branching processes that extended up to 1 cm from the cell body (*Figure 5a–c*; *Figure 5—figure supplement 1*). Cell body areas ranged from 120 to 190 μm² (measured in 2D); this is substantially smaller than the cell body area of dopaminergic neurons, which was measured as 294 ± 4 μm² (n = 10) in our optical mapping experiments (see below). VTA Sst cells had 6–9 processes with 2–4 branching points (*Figure 5a*; *Figure 5—figure supplement 1*).

The structure of Delayed neurons was significantly different from those of the other Sst neurons, as indicated by Sholl plots of neuronal process complexity (*Figure 5c*; n = 20). Indeed, Delayed neurons were different from ADP and HFF neurons in their number of processes, branching points and cell body area (*Figure 5a*); the latter is in agreement with our measurements of cell capacitance (*Figure 3—figure supplement 2*). Staining of filled neurons with anti-Th antibody revealed another unique feature of the

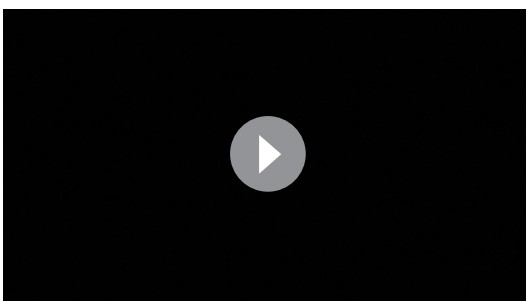

**Video 1.** https://vimeo.com/422563851 A video abstract illustrates three subtypes of somatostatin neurons, described in the article, with their original morphology (traced with neurobiotin with further 3D reconstruction), firing patterns (recorded with patch-clamp method). It also shows location of these neurons within ventral tegmental area (VTA), midbrain and the whole mouse brain. *Video has been made by a media artist Nikolai Larin. 3D Mouse brain credit: Allen Institute.*

https://elifesciences.org/articles/59328#video1

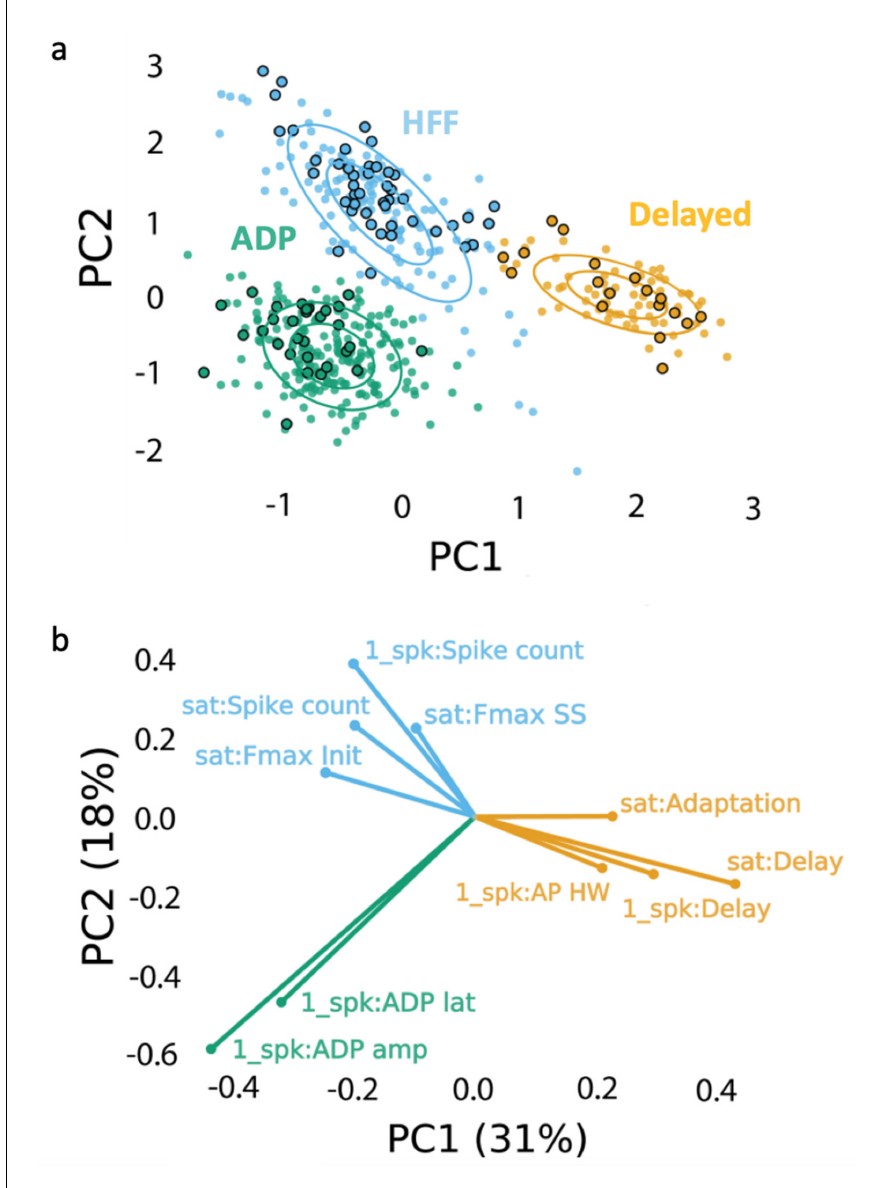

**Figure 3.** Clustering approach for electrophysiological subtyping of the Sst-expressing neurons. Afterdepolarizing 'ADP' subtype is in green, high-frequency firing 'HFF' subtype in light blue, and 'Delayed' subtype in yellow. (**a**) Scatter plot of the Sst-neuron subtypes depicting the results of PCA (PC1 and PC2 as 'x' and 'y' axes, respectively) and GMM (big circles in corresponding colors). Non-outlined dots denote the cells from juvenile mice (n = 392), while black-outlined dots represent mature neurons from P60-P90 mice (n = 92): mature neurons reproduced the clustering pattern of younger neurons. (**b**) PCA weights of the electrophysiological characteristics that most influenced the clustering (see also *Table 1*). Clustering script and intermediate electrophysiological data to reproduce the results can be downloaded from here: https://version.aalto.fi/gitlab/zubarei1/clustering-for-nagaeva-et.-al.-sst-vta. Supporting data can be found in the Additional files: *Figure 3—figure supplements 1–4*. The online version of this article includes the following figure supplement(s) for figure 3:

**Figure supplement 1.** Bayesian information criterion (BIC) for model selection.

**Figure supplement 2.** Passive membrane properties of VTA Sst-neuron subtypes.

**Figure supplement 3.** Electrophysiological properties of the first action potential (AP) at rheobase level of excitation.

**Figure supplement 4.** Electrophysiological properties of the subtypes at the saturating level of excitation, producing the highest number of action potentials.

Delayed neurons: four out of eight tested cells were Th-positive (*Figure 5—figure supplement 2*), which was not observed in the two other Sst neuron subtypes.

## Gene expression in VTA Sst-neurons

The heterogeneity of VTA neurons has been established by the finding that individual neurons can express dopaminergic, glutamatergic and GABAergic markers in different combinations (*Zhang et al., 2015*). To quantify this diversity, we performed multichannel in situ hybridization (RNAscope) for mRNA of *Sst* and three well-described proteins involved in transport of each neuro-transmitter: *Slc17a6* (glutamate), *Slc32a1* (GABA) and *Slc6a3* (dopamine). We observed four main expression patterns for Sst neurons: 75% were *Slc32a1*-positive, 18% *Slc17a6*-positive, 5% *Slc32a1 +Slc17a6* positive, 1% *Slc6a3*-positive and the remaining 1% included all other possible combinations (*Slc17a6+ Slc6a3*, *Slc32a1+ Slc6a3* and *Slc17a6+Slc32a1+ Slc6a3*) (*Figure 6a–b*).

To see how these patterns of neurotransmitter expression are distributed amongst the three different classes of Sst neurons, we combined single-cell electrophysiology, to identify the Sst neuron type, with subsequent single-cell mRNA sequencing (PatchSeq) (*Cadwell et al., 2016*; *Fuzik et al., 2016*) to examine expression of neurotransmitter marker genes. Of the 69 Sst neurons sequenced, 61/69 passed our quality control criteria at all steps of analysis. Sample sizes for different Sst neuron subtypes were unequal due to the abundance of ADP cells: of the 61 cells, 43 were ADP neurons, 9 HFF neurons and 9 Delayed neurons. All except one evinced Sst gene expression. We then mapped PatchSeq data to single-cell RNA-sequencing data from a midbrain data set, in order to find midbrain neuronal subtypes that optimally matched the neurotransmitter identities of PatchSeq cells. Our reference dataset was acquired by clustering mouse midbrain scRNA-seq data, published earlier (*Saunders et al., 2018*), using Seurat algorithm. 10,187 cells from this midbrain dataset, which

**Table 1.** Electrophysiological properties of the three Sst neuron subtypes in the VTA.

| General | ADP (n = 215) | HFF (n = 92) | Delayed (n = 85) |
|---|---|---|---|
| Input resistance (MΩ) | 783 ± 24 | 835 ± 37 | 928 ± 42 |
| RMP (mV) | −69.9 ± 0.6 | −67.8 ± 0.8 | −64.9 ± 1.0 |
| Cm (pF) | 13.2 ± 0.2 | 12.4 ± 0.3 | 15.0 ± 0.4* |
| Sag | 0.963 ± 0.002 | 0.952 ± 0.002* | 0.961 ± 0.003 |
| 1 st spike at rheobase current | | | |
| Rheobase current (pA) | 18.4 ± 1.0 | 11.9 ± 1.2 | 40.8 ± 4.3* |
| AP threshold (mV) | −31.1 ± 0.3 | −31.7 ± 0.5 | −25.6 ± 0.6 |
| AP amplitude (mV) | 95.2 ± 0.6 | 94.0 ± 0.8 | 87.9 ± 0.8* |
| ADP amplitude (mV) | 13.8 ± 0.5* | no ADP | no ADP |
| 1 st AP delay (ms) | 218 ± 10 | 182 ± 14 | 528 ± 25* |
| AP half-width (ms) | 1.01 ± 0.01* | 0.90 ± 0.02* | 1.46 ± 0.04* |
| AHP amplitude (mV) | 26.3 ± 0.5 | 35.7 ± 0.9* | 25.3 ± 1.5 |
| AP decay (ms) | 2.2 ± 0.1 | 1.84 ± 0.04 | 7.9 ± 0.5* |
| Spike count | 2.1 ± 0.1* | 3.9 ± 0.2* | 1.4 ± 0.1* |
| Saturated level of excitation | | | |
| Saturating current step (pA) | 207 ± 5 | 209 ± 11 | 242 ± 12 |
| Fmax initial (Hz) | 108 ± 3 | 129 ± 5 | 46 ± 15* |
| Fmax steady state (Hz) | 25.6 ± 1.1 | 42.9 ± 3.7 | 21.2 ± 1.4 |
| Adaptation ratio | 0.28 ± 0.02 | 0.32 ± 0.02 | 0.80 ± 0.04* |
| 1 st AP delay (ms) | 24.8 ± 3.4 | 12.2 ± 1.2 | 279.0 ± 20.4* |
| AP half-width (ms) | 1.88 ± 0.04* | 1.6 ± 0.1* | 2.5 ± 0.1* |
| Spike count | 21.6 ± 0.9* | 39.1 ± 3.1* | 11.00 ± 0.8* |

Data are shown as means ± SEM. Asterisks indicate values statistically different from two others (p<0.05). Row colors indicate important features, which influenced unsupervised clustering (*Figure 3a–b*).

passed the quality control filter (see Materials and methods), were distributed in 19 clusters, six of which were neuronal (*Figure 6—figure supplement 1*). We then mapped our Sst cells onto these six neuronal subtypes using a bootstrapping algorithm (*Muñoz-Manchado et al., 2018*). The bootstrapping algorithm reveals the probability that a cell can be assigned to existing clusters. To our surprise, we observed that several cells could be assigned to more than one cluster with a similar probability. This is likely due to the fact that the reference scRNAseq dataset contained all cells from the midbrain and thus a low number of Sst interneurons from the VTA specifically, resulting in dissolution of the less abundant neuronal subtypes in the major clusters. On the other hand, multiple assignments seem logical, if we take into account the mixed neurotransmitter identity of the Sst neurons demonstrated by RNAscope experiment (see above). Therefore, we focused on the main identity of the clusters which had GABAergic, dopaminergic and glutamatergic identities. Seven PatchSeq cells could not be assigned to any of the clusters, but the remaining 54 Sst neurons were mapped onto three out of the six neuronal clusters that contained Sst-positive cells (*Figure 6c*, *Figure 6—figure supplement 1*). Five out of nine HFF cells were exclusively in the GABAergic cluster (main marker genes were *Gad2, Gad1, Slc32a1, Atp1b1, Snap25, Camta1, Nrxn3, Gata3, Ttc3, Atp1a3, Nsf, Syn2, Sncb* and *Kcnc1*) and two out of nine went also to a glutamatergic cluster (main markers were *Slc17a6, Cacna2d1, Cbln2, Scn2a1, Shox2* and *Adcyap1*), thus being both GABAergic and glutamatergic. Eight out of 43 ADP neurons were assigned exclusively to the GABAergic cluster, whereas most of these cells (26/43) went to both GABAergic and glutamatergic clusters. Eight out of nine Delayed neurons mapped onto a dopaminergic cell type (main markers were: *Slc6a3, Slc18a2, Th, Dlk1, Ret, Ddc, Aldh1a1, Cplx1, Cpne7, Pbx1, Syt1, Rab3c, Slc10a4, Cplx2, Erc2, Chrna4, En1, Epha5* and *Chrna6*). Although six out of nine Delayed cells went also to the glutamatergic cluster, only Delayed cells were assigned to the DA cluster. *Figure 6c* summarizes mapping of five acquired molecular subtypes (GABA, GABA+Glu, GABA+Glu+DA, Glu+DA, and DA) on electrophysiological clusters. Because the Delayed cluster included all variants of dopaminergic molecular subtypes, we examined DA marker expression in the PatchSeq data. Delayed neurons expressed higher levels of DA markers, such as *Th, Slc6a3* (dopamine transporter), *Ddc* and *Slc18a2* (VMAT2), compared to two other neuron subtypes (*Figure 6—figure supplement 2*).

## ADP cells are interneurons

Because our sequencing results indicated that most ADP neurons were GABAergic, we next asked whether these neurons serve as interneurons within the VTA. Slices prepared from mice expressing channelrhodopsin-2 (ChR2) exclusively in Sst neurons (Sst-IRES-Cre x Ai32) allowed us to photostimulate Sst neurons with small laser spots, while recording responses in potential postsynaptic target neurons. Selective photostimulation of ADP neurons in the VTA was achieved by identifying an optimal laser power (9 µW) and light flash duration (4 ms) that was able to reliably evoke individual action potentials exclusively in this particular subtype of Sst neuron, as shown in the optical footprint in *Figure 7a*. The action potential firing properties of the two other Sst neuron subtypes prevented us from using this approach: HFF neurons were unable to fire only single action potentials during brief light flashes, due to their burst-firing properties, while brief light flashes did not evoke any action potentials in Delayed neurons, because of their prolonged delay before action potential firing.

Photostimulation of ChR2-expressing ADP neurons (*Figure 7a*) evoked inhibitory postsynaptic currents (IPSCs) in neighboring VTA DA neurons (*Figure 7b–d*). We observed evoked IPSCs in 42% of DA cells examined (11/26 confirmed DA neurons). The dopaminergic identity of the postsynaptic cells was confirmed by the presence of a hyperpolarization-activated inward current ($I_h$) and/or by post-hoc labeling of these cells by anti-Th antibody (*Figure 7d*; *Figure 7—figure supplement 1*). Optically evoked IPSCs were completely blocked by the GABA$_A$ receptor antagonist, gabazine (10 µM, n = 2), indicating that Sst ADP neurons inhibit DA cells via GABA$_A$ receptors (*Figure 7b*).

The spatial organization of local circuits formed between ADP neurons and DA neurons could then be revealed by optogenetic circuit mapping (*Wang et al., 2007*; *Kim et al., 2014*). The 'optical footprint' of the ADP neurons - the region where the laser spot was capable of evoking an AP (*Kim et al., 2014*) - was largely restricted to the somatic region and excluded the axons of these neurons (*Figure 7a*). By scanning the laser spot over the entire field (0.26 mm$^2$) to sequentially excite all presynaptic ADP neurons, local inhibitory circuits could be mapped by correlating the laser spot

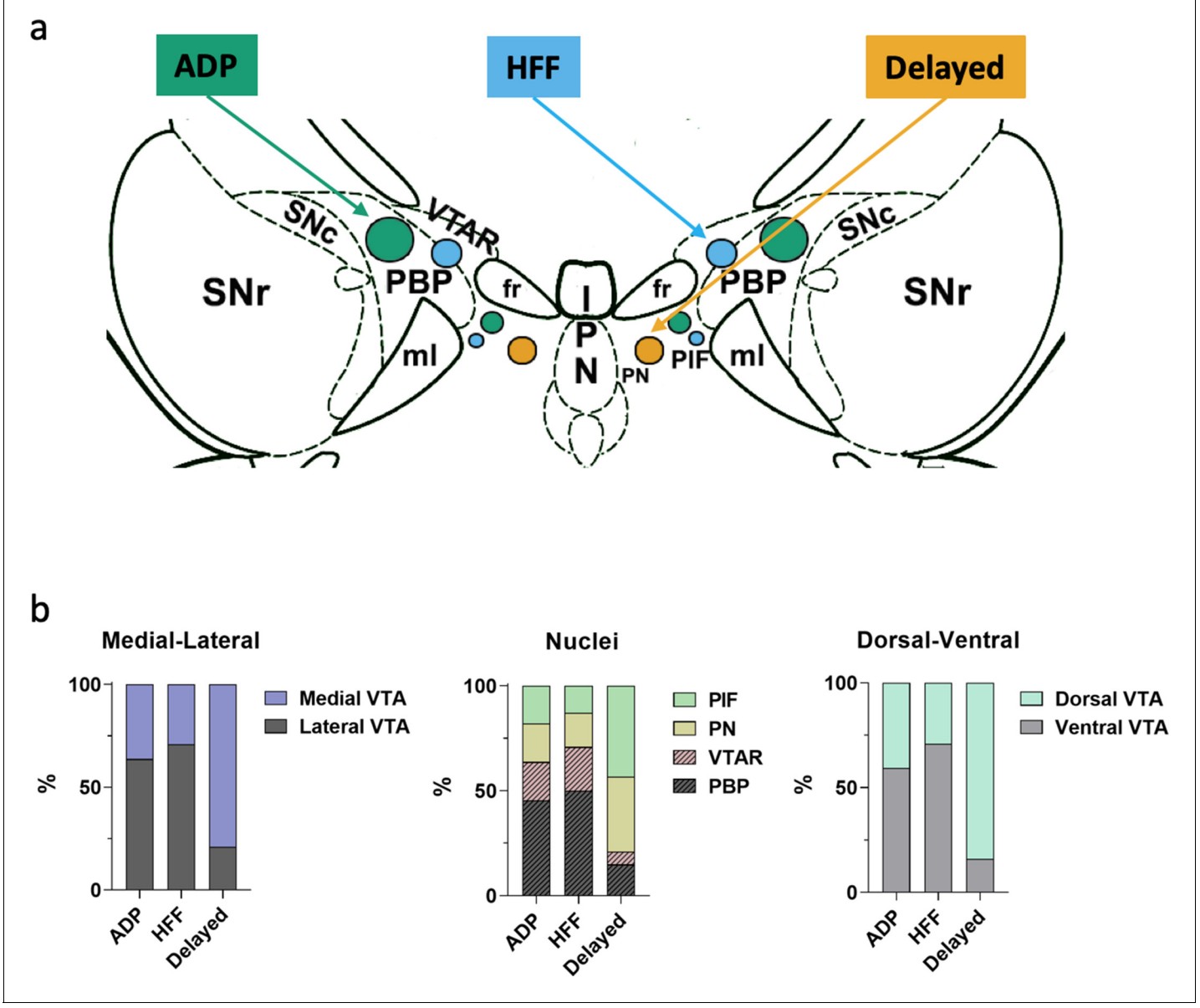

**Figure 4.** Differential localizations of Sst-expressing neuron subtypes within the VTA. (a) Schematic depiction of subtype locations within the VTA horizontal slice (bregma −4.44 mm). Circle sizes represent relative number of cells. SNc, substantia nigra pars compacta; VTAR, ventral tegmental area, rostral part; the other abbreviations as defined in *Figure 1a*. (b) Lateral part of the VTA is represented by PBP and VTA nuclei, and the medial part by PIF and PN nuclei, in accordance with the Mouse Brain Atlas (*Franklin and Paxinos, 2008*). The ventral VTA was defined from −4.72 to −4.56 mm, and the dorsal VTA from −4.44 to −4.28 mm in horizontal plane. The Delayed neurons were preferentially localized in the mediodorsal VTA, while the ADP and HFF neurons lateroventrally.

location with the amplitude of IPSC evoked in postsynaptic DA cells. *Figure 7c,e and f* show examples of input maps, where IPSCs of variable amplitudes (47 ± 3 pA, with a range from 8 to 218 pA, for 170 events from 11 DA cells; *Figure 7—figure supplement 2*) could be evoked by photostimulating ADP neurons over a broad area within the lateral VTA. Many input maps, such as the one in *Figure 7f*, showed more than one discrete cluster of ADP inputs onto a DA cell. This indicates that more than one Sst ADP neuron converges on a postsynaptic DA cell. Further evidence for this comes from quantification of input field size: the median size of these input fields (7400 μm$^2$; n = 11) was much larger than that of the optical footprint of individual ADP cells (640 μm$^2$; n = 20), indicating that more than one presynaptic Sst ADP neuron must contribute to an inhibitory input field. However, because of the high variability in the size of both input fields and optical footprints, we did not

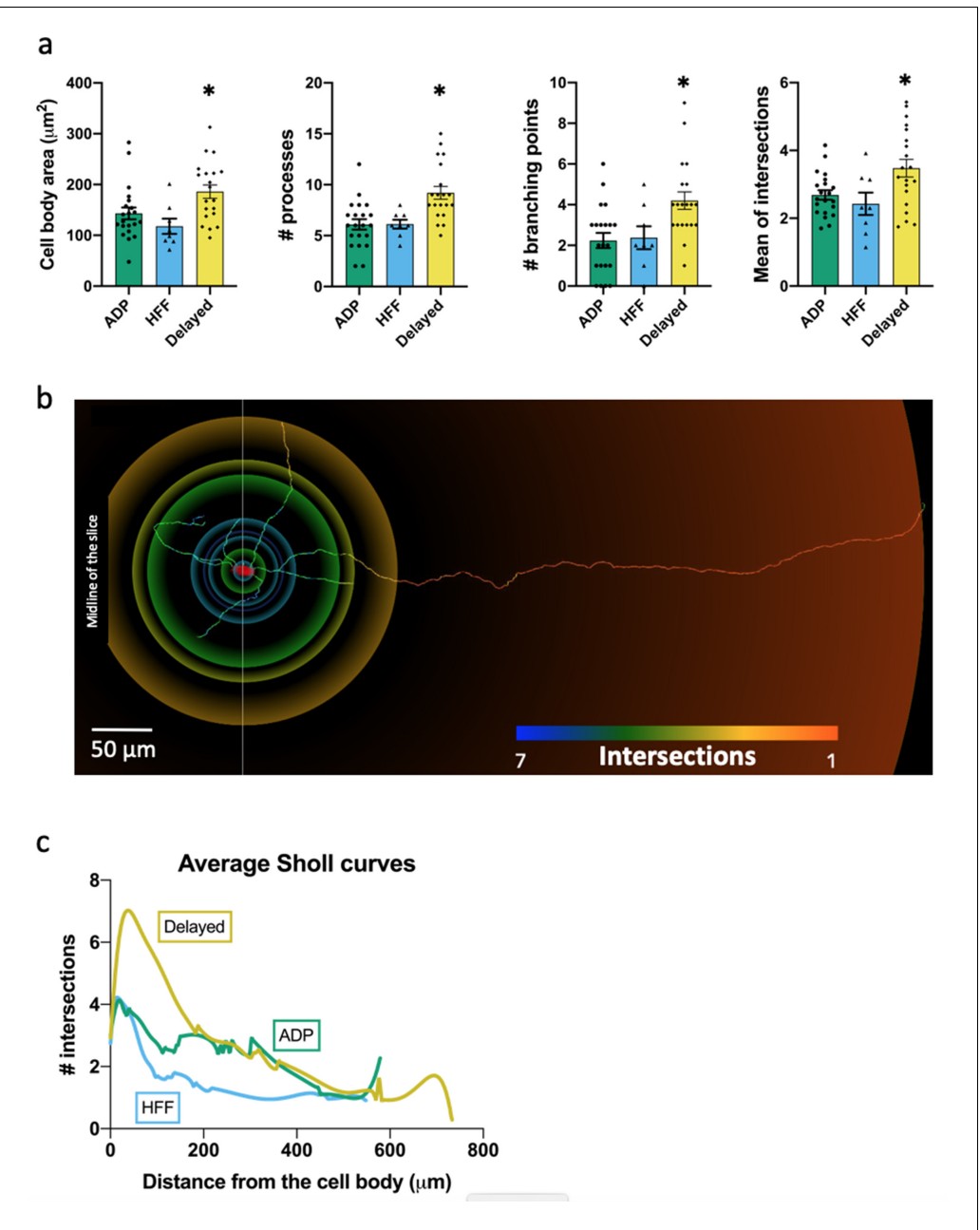

**Figure 5.** Morphological properties of the Sst-expressing neuron subtypes. (a) Significantly different morphological characteristics between the three electrophysiological subtypes. *Mean intersections*, a parameter from Sholl analysis, shows how many times a circle of one radius intersects neuronal processes. All graphs show means ± SEM for ADP neurons (n = 21), HFF neurons (n = 8) and Delayed neurons (n = 20). Asterisks indicate values statistically different from two others (cell body area: F(2, 46)=5.565, p=0.0068; # processes: F(2, 46)=9.745, p=0.0003; # branching points: F(2, 46)=6.974, p=0.0023; mean intersections: F(2, 46)=5.280, p=0.0086). (b) Example of a Delayed neuron morphology with color-coded circles, produced by Sholl analysis: from red to blue for minimum to maximum number of intersections, respectively. White line crossing the cell body corresponds to the direction of the sagittal plane and 'midline' indicates center of the slice close to the interpeduncular nuclei (IPN, see also *Figure 4a*). (c) Average Sholl curves for the electrophysiological subtypes, showing that Delayed neurons had the largest number of intersections within 100 μm from the cell body. Morphology of the traced neurons and individual Sholl curves can be found in Additional files: *Figure 5—source datas 1–2*. *Figure 5—figure supplements 1–2*.

The online version of this article includes the following source data and figure supplement(s) for figure 5:

**Source data 1.** raw data for *Figure 5a*.

**Source data 2.** File Morphology_Source.zip: source files for morphology and location.

**Figure supplement 1.** Examples of VTA Sst-neuron morphology.

**Figure supplement 2.** An example of a neurobiotin-filled (NB+) Delayed neuron with positive Th staining.

attempt to precisely calculate the number of presynaptic ADP cells converging upon a postsynaptic DA cell (*Kim et al., 2014*). In conclusion, our optogenetic mapping experiments indicate that DA neurons in the VTA are inhibited by multiple ADP neurons, establishing ADP cells as inhibitory interneurons that converge upon postsynaptic DA cells.

## Discussion

Single-cell techniques allow multimodal interrogation of cellular properties and have been used for analyses of both known and unknown neuronal populations (*Fuzik et al., 2016*; *Muñoz-Manchado et al., 2018*; *Gouwens et al., 2019*). Here, we combined the patch-clamp method with anatomical, molecular and optogenetic approaches to characterize a previously unknown population of Sst-expressing neurons in the mouse ventral tegmental area. Recording of the intrinsic electrical properties of 392 Sst neurons, in conjunction with clustering algorithms, revealed three major subtypes of Sst neurons in the VTA. These three types of cells - ADP, HFF and Delayed neurons - differed not only in their firing properties, but also in their location within the VTA. Identifying these Sst neuron subtypes in the VTA will enable future efforts to target and manipulate these cells. We have also provided a demonstration of the utility of this classification scheme by selectively photostimulating ADP cells to reveal that they are inhibitory interneurons.

Distinct anatomical locations of Sst cells within the VTA may indicate different physiological functions: lateral and medial Sst neuron clusters co-localize with previously described DA neuron populations (*Lammel et al., 2011*). Lateral DA neurons project predominantly to the lateral shell of the NAc and respond to reward-related events; in contrast, medial-posterior DA neurons project to the medial prefrontal cortex and medial shell of the NAc and undergo plastic changes after aversive stimuli. It is possible that Sst ADP neurons, which are predominantly located in the lateral VTA and inhibit neighboring DA neurons, can control the activity of DA neurons in that particular area, thereby modulating reward processing and motivation signals, as previously established for the lateral VTA GABA neurons (*Eshel et al., 2015*). As local interneurons, ADP cells may also be targets for rewarding drugs, such as opioids (*Johnson and North, 1992*), gamma-hydroxybutyrate (*Cruz et al., 2004*) and benzodiazepines and other GABAergic drugs (*Tan et al., 2010*; *Vashchinkina et al., 2014*), which act via a disinhibitory mechanism by inhibiting local VTA GABA neurons. Thus, our results strongly suggest that Sst ADP neurons play an important role in VTA physiology and pharmacological responses.

Limited data are available on the intrinsic electrical properties of VTA GABA neurons, especially in mice (*Steffensen et al., 1998*; *Nelson et al., 2018*); our results provide such information for VTA Sst neurons. The known electrical properties of VTA DA neurons allow comparison with Sst-neuron subtypes (*Neuhoff et al., 2002*; *Hnasko et al., 2012*). VTA Sst neurons had a lower cell capacitance than DA neurons, reflecting the smaller cell size of Sst neurons compared to DA neurons. The mean capacitance of the three subtypes of Sst cells was 14 pF (*Table 1*), compared to the mean of 41 pF for DA neurons (taken from several studies summarized in https://neuroelectro.org/neuron/203/). Higher input resistance (849 vs. 665 MΩ) and larger AP amplitude (92 vs. 66 mV), as well as narrower action potential half-width (1.1 vs. 2.8 ms) and more hyperpolarized resting membrane potential (−67.5 vs. −51.7 mV) were additional distinctive features of Sst neurons in the VTA. Because 75% of Sst neurons are GABAergic (*Figure 6b*), the comparisons above presumably reflect differences between GABA and DA neurons.

Delayed neurons differed strongly from two other subtypes and, in many respects, resemble DA neurons. Indeed, they showed a wide action potential half-width, relatively large cell bodies and more highly branched dendrites as well as Th+ immunoreactivity, which are all common features of DA cells. Moreover, only Delayed neurons fell into the dopaminergic cluster in our PatchSeq results and the subsequent analysis for DA marker expression showed the highest levels of *Th*, *Slc6a3*, *Ddc* and *Slc18a2* (VMAT2) in this particular subtype compared to two others (*Figure 6—figure supplement 2*). The absence of $I_\mathrm{h}$ current in all Sst neuron subtypes (evident as absence of membrane potential sag during hyperpolarization; see *Table 1* and *Figure 2*), including Delayed neurons, does not contradict our suggestion because medial-posterior VTA DA neurons of mice also do not evince this response (*Lammel et al., 2011*; *Hnasko et al., 2012*). Therefore, Delayed Sst neurons may represent a population of Sst-expressing DA neurons in the VTA.

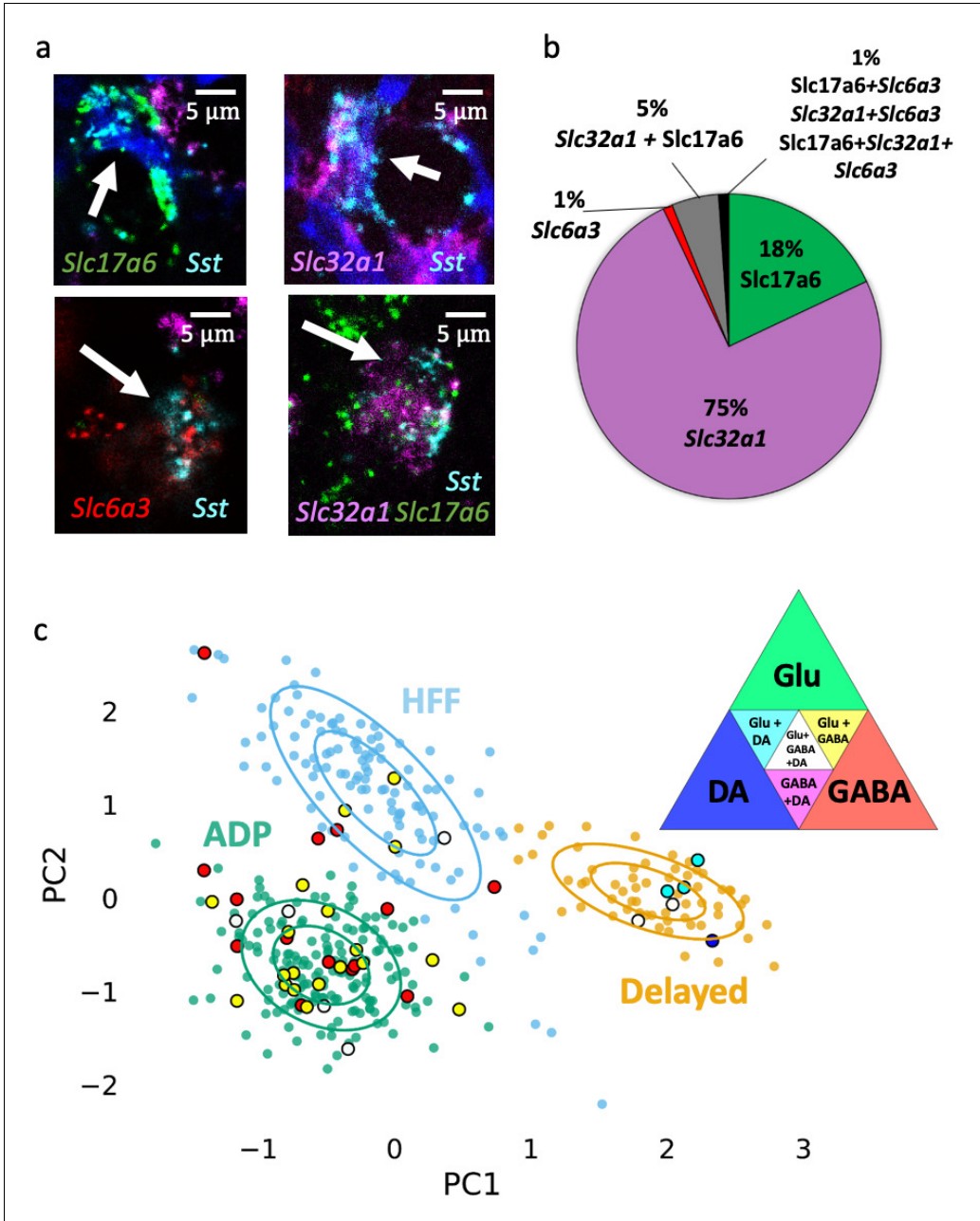

**Figure 6.** Neurochemical phenotypes of VTA Sst neurons according in situ mRNA hybridization and PatchSeq experiments. (a) Example images of multichannel in situ hybridization RNAscope experiments, showing a variety of Sst neuron molecular subtypes, based on *Sst, Slc6a3, Slc32a1 and Slc17a6* mRNA expressions. (b) Proportion of Sst neuron molecular subtypes, as classified and counted for coronal and horizontal sections of the VTA from adult wild-type C57BL/6J mice (n = 8). (c) Molecular subtypes, acquired by alignment of PatchSeq results to a bigger midbrain scRNASeq database, mapped on electrophysiological Sst-neuron clusters (see *Figure 3a*). Triangle in the right corner shows color-coding for the molecular subtypes. ADP and HFF clusters included mostly GABA or Glu+GABA neurons, whereas the Delayed cluster had a spectrum of DA-containing neurons. Supporting data can be found in the Additional files: *Figure 6—source data 1*. *Figure 6—figure supplements 1–4*.

The online version of this article includes the following source data and figure supplement(s) for figure 6:

**Source data 1.** Raw data for *Figure 6b–c*.

**Figure supplement 1.** Seurat clustering of the reference midbrain dataset and selection of the clusters for PatchSeq cells classification.

**Figure supplement 2.** Sst expression in selected clusters 3, 5, 6, 7, 12 and 13.

**Figure supplement 3.** Neuronal marker expression in selected clusters 3, 5, 6, 7, 12 and 13.

**Figure supplement 4.** Expression of dopamine-related genes in different *Sst*-expressing neuronal populations in the VTA, indicating strong expression in Delayed cells.

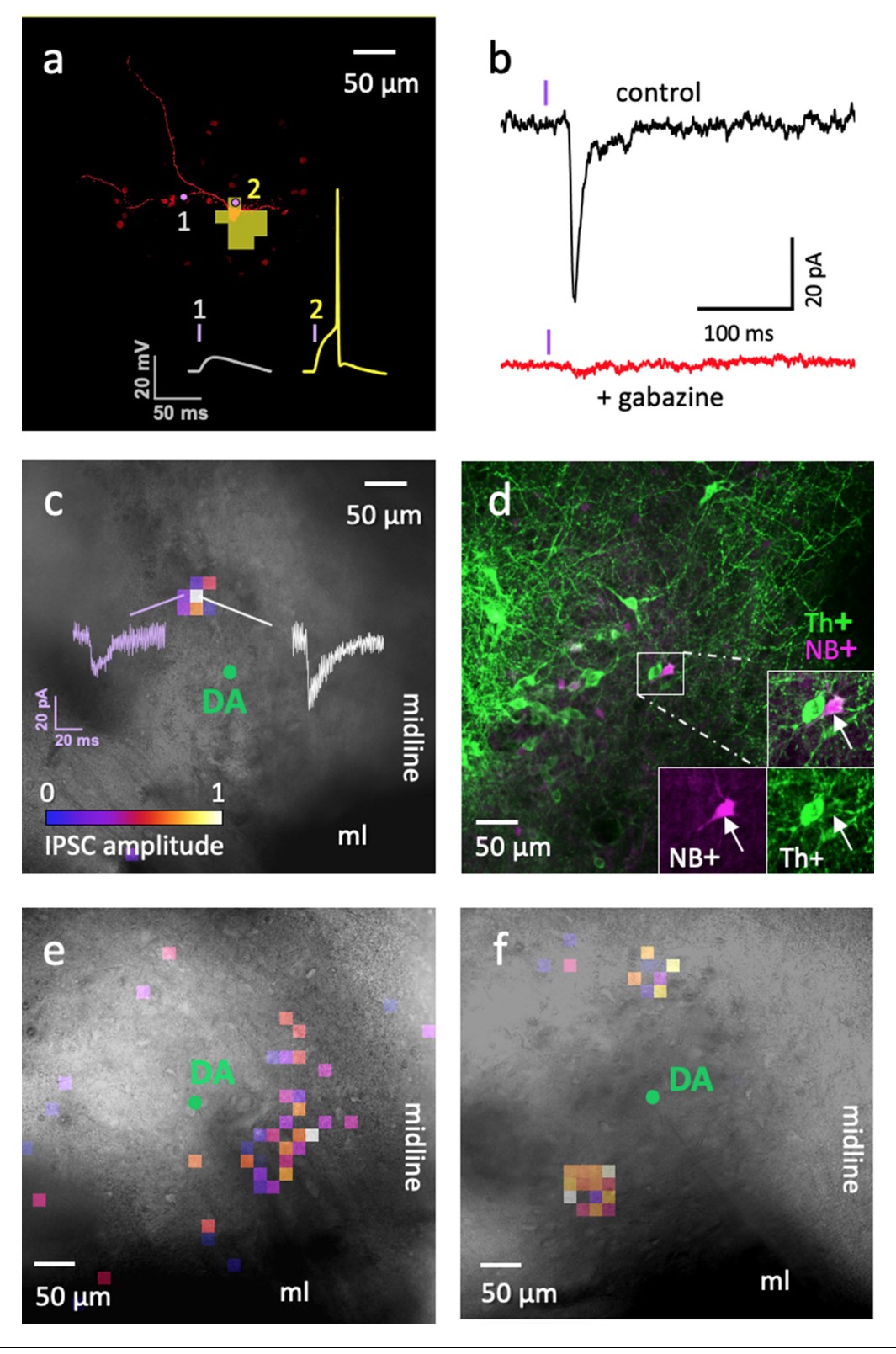

**Figure 7.** Optogenetic circuit mapping: a single VTA DA neuron can be inhibited by more than one ADP Sst-expressing neuron. (a) Representative example of the optical footprint of a Sst neuron stimulated with 405 nm laser spots of 9 µW power. The neuron was filled with Alexa 594 dye through patch pipette. Optical footprint, illustrating the square spots where laser stimulation induced action potential in the ADP subtype of Sst cells, is

*Figure 7 continued*

shown in yellow pixels and located around the cell body. Traces 1 and 2 demonstrate evoked responses, while stimulating the corresponding point around the Sst cell. Action potential (2) was induced by stimulation on the soma, while no action potentials were induced by laser spots on the neurites (1). (**b**) The black trace ('control') shows an IPSC evoked in DA cell due to optical stimulation of a neighboring ADP neuron. The red trace shows an absence of evoked IPSCs in the same cell during application of the GABA$_A$ receptor antagonist gabazine (10 µM). The violet bars indicate moments of the light (405 nm) flash. (**c**) Colored pixels represent input maps from a Sst neuron to a neighboring DA cell (shown as a green circle in the center) at 9 µW laser power setting. Amplitudes of the optically evoked IPSCs were color-coded according to the pseudocolor scale, shown at the bottom in relative units from 0 to 1 (used also in panels **e** and **f**). Actual amplitudes varied between experiments and are here shown in white (corresponds to maximum IPSC amplitude of 40 pA) and purple (corresponds to IPSC amplitude of 20 pA) traces. (**d**) Post-staining of the neurobiotin-filled postsynaptic DA neuron shown in **c**; the cell is labeled with streptavidin 633 (magenta, NB+) and tyrosine hydroxylase antibody (green, Th+). (**e–f**) Two other examples of input maps at 9 µW demonstrate their variability. All circuit maps in **c**, **e** and **f** overlay images of horizontal VTA slices, where recordings were carried out showing location of the DA cells. Maps in **c** and **e** were recorded at bregma −4.44 and that in **f** at bregma −4.56. ml, medial lemniscus; 'midline' indicates the sagittal center line of the horizontal slice. Supporting data can be found in the Additional files: *Figure 7—figure supplements 1–2*. The online version of this article includes the following source data and figure supplement(s) for figure 7:

**Figure supplement 1.** Examples of I$_h$-current test (**a**) and immunohistochemistry (**b**) for confirmation of DA neuron phenotype.

**Figure supplement 2.** Examples of amplitude variations of evoked IPSCs in VTA DA cells.

**Figure supplement 2—source data 1.** Raw data for *Figure 7—figure supplement 2*.

Our experience with the PatchSeq method indicated the need for a more specific VTA GABA scRNAseq dataset. Currently, there are only two whole-midbrain datasets available (*Saunders et al., 2018*; *Zeisel et al., 2018*), both of which were made as part of whole-brain sequencing projects. Therefore, dissection precision and sequencing depth achieved in those studies do not produce sufficient resolution to distinguish different midbrain nuclei. Simultaneous alignment to several clusters during mapping of our PatchSeq data may have resulted from this problem: the whole midbrain dataset includes Sst neurons not only from the VTA, but also from the substantia nigra and the interpeduncular nucleus, which together contain at least three times more Sst neurons than the VTA (*Kim et al., 2017*). Therefore, smaller but distinct VTA Sst populations could be lost within larger DA, Glu or GABA neuron clusters. Nevertheless, our bootstrapping algorithm could establish logical assignments to GABA, Glu and DA clusters.

The PatchSeq method uses aspirated cell content samples, making it possible for contamination to occur as the recording pipette passes through other cells and processes (*Tripathy et al., 2018*). However, our main conclusions from the PatchSeq data regarding the mixed neurochemical phenotypes of Sst neurons are in agreement with our in situ hybridization results (*Figure 6*). Our conclusion regarding the DAergic nature of the Delayed neurons was also supported by independent evidence, such as their immunohistochemical staining for Th, their morphology and their electrophysiological parameters (see above). The exclusive prevalence of DA markers in Delayed neurons, but not in the other 52 neurons, also provides an indication of the quality of our PatchSeq data.

Cortical GABA neurons and midbrain GABA neurons originate from different brain areas and precursor cells and also have different local environments (*Xu et al., 2004*; *Achim et al., 2012*). Therefore, it was important to check out whether the neocortical Sst-GABA neuron marker genes (*Tasic et al., 2018*) would be expressed in the Sst neurons of the midbrain dataset (*Saunders et al., 2018*) and in our PatchSeq-sampled Sst neurons. To be able to compare the three datasets of different brain regions and collected with different sequencing technologies, we extracted all *Sst*-expressing neurons and calculated means of gene expression for these cells and used *Sst* gene expression value set at 1000 as the normalizing factor between the brain regions. About two thirds of marker genes identified in *Tasic et al., 2018* for neocortical Sst neurons were expressed in the midbrain dataset or in the PatchSeq samples (*Figure 8*). *Tac2,* a marker gene of two different subtypes of neocortical Sst neurons (*Tac2-Myh4* and *Tac2-Tacstd2*), as well as some cortical interneuron markers

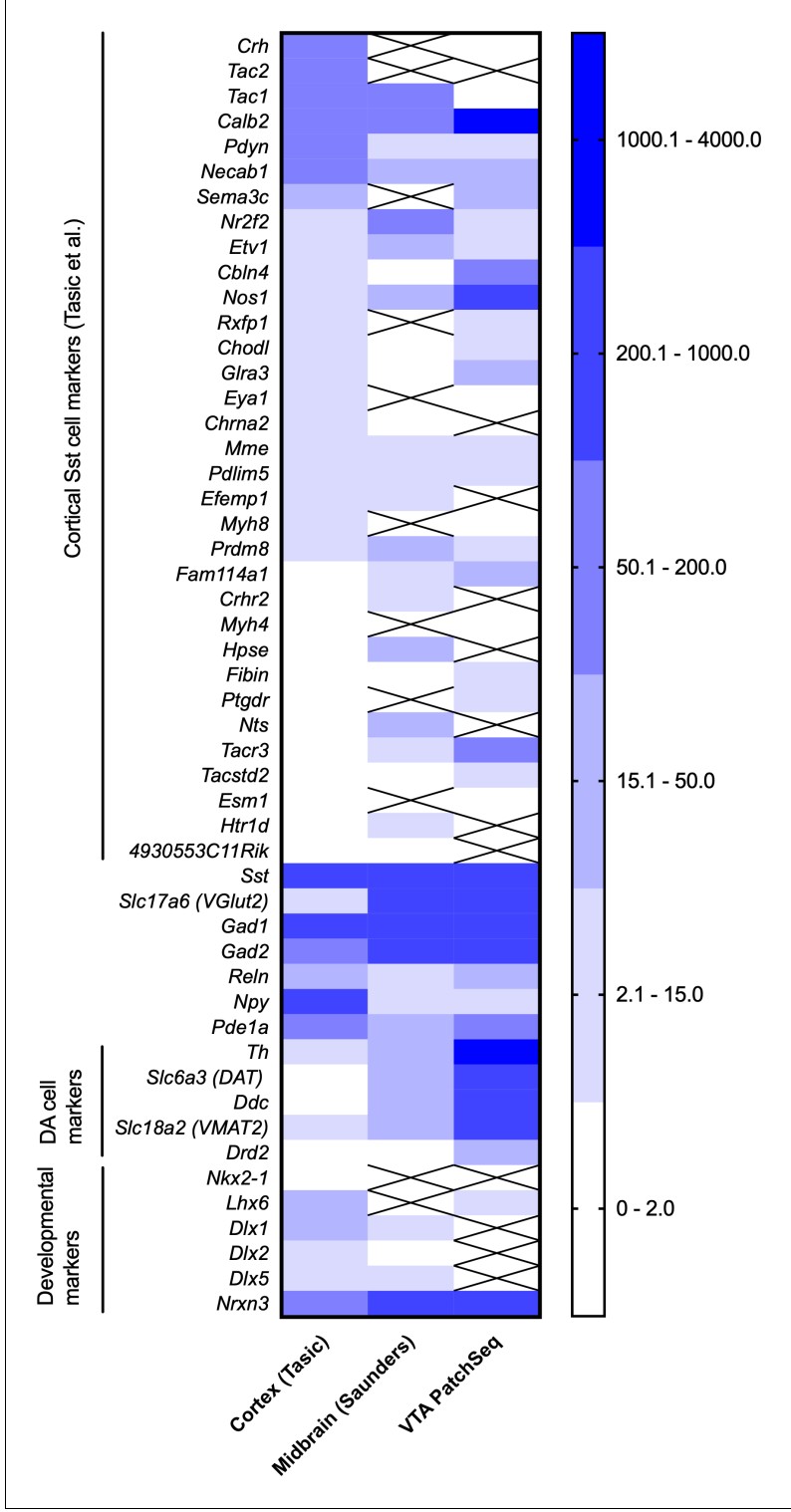

**Figure 8.** Comparison of expression of genes of interest in neocortical Sst neurons (*Tasic et al., 2018*), Sst neurons of the midbrain (*Saunders et al., 2018*) and in the PatchSeq cells of the present study. The three sets were normalized by setting the *Sst* expression to 1000. Crossed out means 'no expression' in the dataset.

*Crh*, *Npy* and *Pdyn* genes (*Sugino et al., 2006*), showed a high expression in the neocortical dataset, but were absent or lowly expressed in the midbrain and PatchSeq datasets. Interestingly, *Chodl*, a gene expressed in the long-range Sst projection neurons of the neocortex (*Tasic et al., 2016*), and *Nr2f2* and *Necab1* showed very low levels of expression in the PatchSeq and midbrain datasets, while *Calb2* and *Nos1* were more pronounced in the PatchSeq cells. On the other hand, Sst neurons in VTA and midbrain Sst neurons had higher expression of glutamatergic (*Slc17a6*) and dopaminergic markers (*Th, Slc6a3, Ddc*) than those in the neocortex, suggesting a low level of combinatorial neurotransmitter phenotypes in the cortex as compared to the midbrain within the Sst neurons. Most of the developmental marker genes (*Chen et al., 2017*; *Hu et al., 2017*) were poorly expressed in PatchSeq and midbrain Sst neurons, but interestingly, *Nrxn3*, a gene for neurexin 3, a presynaptic terminal protein stabilizing glutamate synapses (*Dai et al., 2019*) and serving as a candidate susceptibility gene for neurodevelopmental disorders such as autism (*Wang et al., 2018*), showed robust expression in PatchSeq and midbrain Sst neurons (*Figure 8*), in line with the expression of glutamatergic markers. These comparisons suggest that the VTA and midbrain Sst neurons clearly differ from the more heterogeneous neocortical Sst neurons, indicating a smaller number of molecular subtypes of neurons in the midbrain than cortex.

By using robust optogenetic circuit mapping, we were able to show that Sst ADP cells function as local interneurons, with about half of the VTA DA neurons receiving $GABA_A$ receptor-mediated IPSCs from one or more ADP cells. However, the present data do not exclude the possibility that any of the three Sst neuron subtypes are projection neurons. Indeed, our preliminary viral-tracing experiments in VTA-injected Sst-Cre mice reveal fibers in the forebrain regions, such as the amygdala, bed nuclei of stria terminalis and paraventricular thalamic nuclei (Nagaeva, Elsilä, unpublished results). Further work will be needed to establish relevant marker genes for Sst neuron subtypes that would enable improved characterization of Sst neurons.

Our results reveal robust heterogeneity of the Sst-expressing neurons within the VTA, extending the current classification of VTA GABA neurons (*Bouarab et al., 2019*; *Paul et al., 2019*). Although specific VTA neuron populations have been recently assigned to specific functions/behaviors, such as VTA GABA neurons to sleep induction and VTA Glu/NOS1 neurons to wake-activity regulation (*Yu et al., 2019*) and NeuroD-expressing DA/Glu neurons to stimulant reward (*Bimpisidis et al., 2019*), our present data on combinatorial Sst neuron populations suggest complex, multidirectional effects on the activity of target cells (*Root et al., 2014*; *Taylor et al., 2014*). Further research will be needed to define the specific functions of VTA Sst-expressing neurons in the regulation and signaling of the midbrain circuitry. The classification scheme that we have established for these neurons will greatly facilitate such studies.

## Materials and methods

### Key resources table

| Reagent type (species) or resource | Designation | Source or reference | Identifiers | Additional information |
|---|---|---|---|---|
| Genetic reagent (*M. musculus*) | Sst[tm2.1(cre)Zjh]/J; Sst-IRES-Cre | The Jackson Laboratory | RRID:IMSR_JAX:013044 | |
| Genetic reagent (*M. musculus*) | B6.Cg-Gt(ROSA) 26Sor[tm14(CAG-tdTomato)Hze]/J; Ai14 | The Jackson Laboratory | RRID:IMSR_JAX:007914 | |
| Genetic reagent (*M. musculus*) | B6;129S-Gt(ROSA) 26Sor[tm32(CAG-COP4* H134R/EYFP)Hze]/J | The Jackson Laboratory | RRID:IMSR_JAX:012569 | |
| Strain, strain background (*M. musculus*) | C57BL/6J | The Jackson Laboratory | 000664; RRID:IMSR_JAX:000664 | |
| Antibody | Anti-tyrosine hydroxylase (Rabbit polyclonal) | Sigma-Aldrich | Cat# AB152, RRID:AB_390204 | IHC: 1:100 |

*Continued on next page*

*Continued*

| Reagent type (species) or resource | Designation | Source or reference | Identifiers | Additional information |
|---|---|---|---|---|
| Antibody | Anti-rabbit IgG H and L Alexa Fluor 488 (Goat polyclonal) | Invitrogen | Cat# A-11008, RRID:AB_143165 | IHC: 1:1000 |
| Peptide, recombinant protein | Streptavidin, Alexa Fluor 633 conjugate | Thermo Fisher | Cat# S-21375, RRID:AB_2313500 | (1:1000) |
| Software, algorithm | Zeiss ZEN 2 (Blue) | Zeiss | RRID:SCR_013672 | |
| Software, algorithm | Leica Application Suite X | Leica | RRID:SCR_013673 | |
| Software, algorithm | Fiji Image J | Fiji https://imagej.net/Fiji/Downloads | RRID:SCR_003070; | Version 1.51; Plugins: Neurite Tracer |
| Software, algorithm | MATLAB | Mathworks (https://www.mathworks.com/) | RRID:SCR_001622 | Version R2018b; custom script for electrophysiological analysis https://github.com/zubara/fffpa |
| Software, algorithm | pClamp | Molecular devices | RRID:SCR_011323 | Clampex 8.2 and 10, Clampfit 10.7 |
| Software, algorithm | R Project for Statistical Computing | https://www.r-project.org | RRID:SCR_001905 | Version 3.5; Seurat 2.3.4 https://satijalab.org/seurat/install.html and EWCE (https://github.com/NathanSkene/EWCE) packages for scRNAseq analysis |
| Software, algorithm | Python Programming Language | https://www.python.org | RRID:SCR_008394 | Version 3.6; custom script for electrophysiological clustering analysis https://version.aalto.fi/gitlab/zubarei1/clustering-for-nagaeva-et.-al.-sst-vta |
| Software, algorithm | CSC Chipster | CSC – IT center for science LTD. (csc.fi) | https://chipster.csc.fi | HISAT2 and HTSeq implementations |
| Software, algorithm | Graphpad Prism | Graphpad | RRID:SCR_002798 | Version 8.1 |
| Commercial assay or kit | RNAScope probe Mm-*Slc6a3* | Advanced Cell Diagnostics | ACD: 315441 | |
| Commercial assay or kit | RNAScope probe Mm-*Slc17a6* | Advanced Cell Diagnostics | ACD: 319171-C2 | |
| Commercial assay or kit | RNAScope probe Mm-*Sst* | Advanced Cell Diagnostics | ACD: 404638 | |
| Commercial assay or kit | RNAScope probe Mm-*Slc32a1* | Advanced Cell Diagnostics | ACD: 319191 | |

## Animals

For image analysis and targeted manipulation of somatostatin (Sst) -positive neurons we used heterozygous male and female mice resulted from cross-breeding of Sst-IRES-Cre (Sst[tm2.1(cre)Zjh]/J) strain with Ai14 tdTomato reporter strain (B6.Cg-Gt(ROSA)26Sor[tm14(CAG-tdTomato)Hze]/J), or with Ai32 reporter strain (B6;129S-Gt(ROSA)26Sor[tm32(CAG-COP4*H134R/EYFP)Hze]/J) for optical mapping experiments (The Jackson Laboratory, Bar Harbor, ME). Animals were group-housed in IVC-cages under 12:12 hr light/dark cycle with ad libitum access to food and water. Animal experiments were authorized by the National Animal Experiment Board in Finland (Eläinkoelautakunta, ELLA; Permit Number: ESAVI/1172/04.10.07/2018) and Institutional Animal Care and Use Committee in Singapore ( NTU-IACUC).

## Whole VTA cell counting

Three 4-month-old Sst-tdTomato mice of both sexes were anesthetized with pentobarbital (200 mg/kg i.p., Mebunat, Orion Pharma, Espoo, Finland) and perfused transcardially with cold 1xPBS solution followed by 4% paraformaldehyde solution. After overnight in the same fixative solution, brains were transferred to 30% sucrose until sinking (at least 48 hr). The brains were then frozen on dry ice and stored at −80°C until sectioned. Sequential 40-μm-thick sections from bregma −2.8 to −3.8 mm were carefully collected to ensure that we could later reconstruct the brain from a series of images obtained with a microscope. For the identification of VTA region and DA cell counting staining with anti-tyrosine hydroxylase (Th) antibody was performed (primary antibody AB152 with 1:100 dilution, Sigma-Aldrich, St. Louis, MI; secondary antibody Alexa Fluor 488, ab150077, with 1:1000 dilution, A11008, Invitrogen, Carlsbad, CA). Tiled images were then obtained using an upright wide field epifluorescence microscope Zeiss Axio Imager Z2. Magnification of 10x with z-stack was found necessary to be able to analyze the entire depth of the section. In some cases, magnifications of 20x and 40x were used for better identification of co-stained cells in the counting phase.

Image analysis was done with Image J software (Wayne Rasband, NIH, MD) and manual counting. The area of the VTA was defined according to Mouse Brain Atlas (*Franklin and Paxinos, 2008*) and positive Th staining, and the borders were drawn on images. All Sst- and Th-positive full-bodied cells were marked, counted and reported for all VTA nuclei. Total estimated counts of positive cells were based on nine sections (representing 40% of the whole VTA) from each three mice. The final number of cells in one mouse was calculated according to the formula: (number of counted cells/40) x 100.

## Multichannel in situ hybridization

Brains of 4-month-old wild-type C57BL/6J mice were dissected and frozen on dry ice, then stored at −80°C. Two female and two male brains were cut into 12-μm-thick coronal sections (bregma −3.07, −3.27, −3.51, −3.63, −3.79 and −3.87 mm), representing 9% of the whole VTA. In addition, other two female and two male brains were cut into 12-μm-thick horizontal sections (bregma −4.72, −4.56, −4.44 and −4.28 mm). All sections were cut with a cryostat (CM3050S, Leica Biosystems, Wetzlar, Germany), mounted on superfrost plus glasses (Menzel-Gläser, Braunschweig, Germany) and stored at −80°C. RNAscope was carried out according to manufacturer's protocol (Advanced Cell Diagnostics, Biotechne, Newark, NJ). Following probes were used: Mm-*Slc6a3* (dopamine transporter), Mm-*Slc17a6* (vesicular glutamate transporter 2), Mm-*Sst* (*Sst*) and Mm-*Slc32a1* (vesicular GABA transporter). For detection OPAL 570, 520, 620 and 690 fluorophores (Perkin Elmer, Waltham, MA) were used. Imaging of the results was done on white laser confocal microscope Leica TCS SP8 X (Leica, Wetzlar, Germany). Analysis of co-labeling and counting of the positive cells were done manually. The final number of cells in one mouse was calculated according to the formula: (number of counted cells/9) x 100.

## Electrophysiology

Electrophysiological experiments were performed with juvenile P17-P23 and adult P55-P90 mice of both sexes. After decapitation, brains of juvenile mice were immediately transferred to carbogen-oxygenated (95% $O_2$ + 5% $CO_2$) ice-cold sucrose-based cutting solution (containing in mM): 60 NaCl, 2 KCl, 8 $MgCl_2$, 0.3 $CaCl_2$, 1.25 $NaH_2PO_4$, 30 $NaHCO_3$, 10 D-glucose and 140 sucrose. Horizontal 225-μm-thick midbrain slices were prepared in the same solution using a vibratome HM650V (Thermo Scientific, Waltham, MA) and then transferred to constantly oxygenated artificial cerebrospinal fluid (aCSF) (containing in mM): 126 NaCl, 1.6 KCl, 1.2 $MgCl_2$, 1.2 $NaH_2PO_4$, 18 $NaHCO_3$, 2.5 $CaCl_2$ and 11 D-glucose (*Heikkinen et al., 2009*). After 15 min at 33°C, slices were incubated at room temperature until the end of experiment (~6 hr). For obtaining slices from adult brains, the mice were first anesthetized with a single intraperitoneal injection of pentobarbital (Mebunat 200 mg/kg) and transcardially perfused with ice-cold N-methyl-D-glucamine (NMDG)-based solution (containing in mM): 93 NMDG, 2.5 KCl, 1.2 $NaH_2PO_4$, 30 $NaHCO_3$, 20 HEPES, 25 D-glucose, 5 Na-ascorbate, two thiourea, 3 Na-pyruvate, 10 $MgSO_4$, 0.5 $CaCl_2$, pH adjusted to 7.35 with 10 N HCl (*Zhao et al., 2011*). After 15 min incubation in the same NMDG solution at 33°C, slices were transferred to room temperature until the end of experiment into constantly oxygenated HEPES-aCSF solution (containing in mM): 92 NaCl, 2.5 KCl, 1.2 $NaH_2PO_4$, 30 $NaHCO_3$, 20 HEPES, 25 D-glucose,

5 Na-ascorbate, two thiourea, 3 Na-pyruvate, 10 MgSO$_4$, 0.5 CaCl$_2$, with pH adjusted to 7.35 with 10 N NaOH.

After 45 min of recovery, slices were transferred into a recording chamber with continuous perfusion with aCSF at 33˚C. Red fluorescence of Sst-positive neurons was detected with an epifluorescence microscope BX51WI (Olympus, Tokyo, Japan) and a CCD monochrome camera XC-E150 (Sony, Tokyo, Japan). Whole-cell current-clamp recordings were made using 3–5 MΩ borosilicate glass electrodes filled with intracellular solution (IS) (containing in mM): 130 K-gluconate, 6 NaCl, 10 HEPES, 0.5 EGTA, 4 Na$_2$-ATP, 0.35 Na-GTP, 8 Na$_2$-phosphocreatine (pH adjusted to 7.2 with KOH, osmolarity ~285 mOsm) (*Fuzik et al., 2016*). For morphological reconstruction experiments, the IS was supplemented with 3 mg/ml biocytin (Tocris, Bristol, UK) or 1.5 mg/ml neurobiotin (Vector Laboratories, Burlingame, CA), and for PatchSeq experiments with 1 U/µL TaKaRa RNAse inhibitor (Takara, Shiga, Japan) immediately before the experiment. Liquid junction potential (+12 mV for all modifications of the IS) was not corrected during recordings.

All electrophysiological experiments were made with an Axopatch 200B amplifier (Molecular Devices, San Jose, CA) filtered at 2 kHz, and recorded with 10 kHz sampling rate using pClamp 8.2 software (Molecular Devices). After achieving whole-cell configuration in voltage-clamp mode (−70 mV), cell capacitance was determined by the 'Membrane Test' feature of the Clampfit software and the amplifier was then switched to current-clamp mode with no extra current applied, allowing cells to be at their own resting membrane potential (RMP) during recordings. Depolarized cells with RMP higher than −50 mV were excluded. For measuring passive and active membrane properties, neurons were injected with 800 ms current steps increasing from −100 to +600 pA with 10 pA increments. After this short protocol, either the cells were filled with biocytin/neurobiotin for 15 min or cell contents were immediately extracted with negative pressure for further RNA analysis. All recordings were performed with intact GABAergic and glutamatergic transmission (i.e. no pharmacological agents were added to the aCSF).

## Firing pattern analysis

For definitions, please see the Methods Table in Appendix 1. Automated analyses of current-clamp recordings were made by custom-written MatLab script. Beforehand, all files were manually checked to exclude possible artefacts. RMP (in mV) was measured at 0 pA in current-clamp mode and calculated as an average baseline of all sweeps (sweep = a single trace of voltage response to injected current step) in a record. Cells with standard deviation of RMP more than ±6 mV were excluded from further analysis. Input resistance (in MΩ) was calculated as a regression coefficient between injected current steps and voltage responses, measured over the last 50 ms of each sweep without any action potentials (AP). 'Sag' was defined as the ratio between steady state voltage mean in the last 50 ms of −100 mV sweep and the most negative voltage peak. Membrane time constant (τ, in ms) was calculated by fitting the −100 mV sweep with a single exponential.

## Parameters calculated from the sweep with the first action potential

Rheobase current step (in pA) was defined as the amplitude of injected current step on which the first AP appeared. AP threshold (in mV) was defined as the voltage point after which voltage grew faster than 10 mV/ms. Latency to the first spike (in ms) was the time interval between the beginning of the current step and that of the AP. AP amplitude (in mV) was the difference between the AP threshold and its positive peak. AP half-width was the width of the AP measured at the halfway from AP threshold to AP peak. Afterhyperpolarization (AHP) amplitude (in mV) was the difference between the AP threshold and the most negative membrane potential reached during afterhyperpolarization. AP decay time (in ms) was the time from the AP peak to the AHP peak. Afterdepolarization (ADP) amplitude (in mV) was the difference between the AHP peak and the most positive membrane voltage during the fast repolarization phase. ADP latency (in ms) was the interval between the AHP and ADP peaks.

## Parameters calculated from the sweep with the highest number of APs (saturated level)

Rheobase current and the latency to the first spike were defined in the same manner as for the sweep with the first AP. Interspike interval (ISI, in ms) was the interval between peaks of two

neighboring spikes. Initial maximal frequency (Fmax init, in Hz) was the inverse of the shortest ISI among the first three spikes. Steady-state maximal frequency (Fmax steady-state) was the inverse of the average mean of the last four ISIs. Adaptation ratio (dimensionless) was the ratio of Fmax steady-state to Fmax initial.

## Clustering using electrophysiological features

All features extracted from current-clamp recordings (25 overall) were used for further clustering analysis. Non-normally distributed features, corresponding to spike counts or latency intervals were log-transformed. After that cells containing parameter estimates lying outside of 5 inter-quartile (IQR = Q3 – Q1) ranges from the median value, were considered outliers and discarded from the analysis. This procedure resulted in discarding a total of 10 samples (eight juvenile and two adult). The remaining data were scaled to the interval [0,1] by subtracting the minimal value followed by division by the maximal value across samples (min-max scaling).

Optimal model selection was performed in two consecutive steps using Bayesian Information Criterion (BIC), a widely used model comparison criterion balancing the goodness-of-fit (based on likelihood function) against the model complexity (or the number of free parameters).

First, BIC was used to identify an optimal number of principal components (PCs) to obtain low-dimensional representation of the data. BIC was estimated for principal component analysis (PCA) decompositions with number of components ranging from 1 to 9 using a fivefold cross-validation. This analysis indicated that PCA decomposition with two principal components accounting for 31.8% (PC1) and 13.9% (PC2) of the variance in the original data provided the best (minimal) BIC estimates. After that, the data were projected onto the twodimensional principal component space and was fit with Gaussian Mixture Models, as implemented in Scikit-Learn Python package (*Pedregosa et al., 2011*) (http://jmlr.csail.mit.edu/papers/v12/pedregosa11a.html), with number of clusters ranging from 1 to 7. Similarly, the best model was chosen based on the minimal average BIC estimated with fivefold cross-validation.

## Morphological reconstruction and analysis

After patch-clamp recording and filling with biocytin/neurobiotin tracer, horizontal midbrain slices were fixed in 4% paraformaldehyde (PFA) solution for 24 hr at 4℃. All following staining procedures were done as described previously (*Marx et al., 2012*; *Mao et al., 2019*), depending on the tracer. Biocytin-filled, DAB (3,3'-Diaminobenzidine) stained neurons (N = 27) were imaged with Zeiss Axio Imager Z2 microscope (Zeiss AG, Oberkochen, Germany) in transmitted light mode. Neurobiotin-filled, Streptavidin-Alexa 633 stained neurons (n = 22) were imaged with multiphoton Zeiss LSM 7 MP system. 3D morphology reconstruction was done with Simple Neurite Tracer plugin (https://imagej.net/Simple_Neurite_Tracer) in Image J software, followed by inbuilt Sholl analysis instrument.

## RNA extraction and lysis

For PatchSeq experiments, juvenile animals were anesthetized and perfused with 30 ml of ice-cold sucrose-based cutting solution before decapitation. All other slicing procedures were as described above (see 'Electrophysiology' section). Set-up and instruments were cleaned up with 70% ethanol, Milli-Q water and RNAse away surface decontaminant (Thermo Scientific). Glass electrodes were filled with a minimum amount of IS (~1 µl). After extraction, cell content was ejected onto 1.1 µl drop of lysis buffer (0.1% Triton X-100, 2 U/µl TaKaRa RNAse inhibitor, 0.5 µM C1-P1-T31 (5'-AAT GAT ACG GCG ACC ACC GAT CGT$_{31}$−3'), 11.5 mM dithiothreitol and 2.3 mM of dNTP mix) placed on the wall of cold RNAse-free PCR tube, spun down and immediately frozen in dry ice. All samples were stored at −80℃ until the reverse transcription step.

## Reverse transcription and library preparation

These steps were carried out as described previously (*Picelli et al., 2014*; *Fuzik et al., 2016*). Briefly, before reverse transcription (RT), samples were thawed at 72℃ for 3 min. After that, 2 µl of RT mix (1x SuperScript II first-strand buffer, 18 U/µl SuperScript II reverse transcriptase (Life Technologies, Carlsbad, CA), 3.6 µM C1-P1-RNA-TSO (5'-AAU GAU ACG GCG ACC ACC GAU NNN NNG GG −3'), 6 mM MgCl$_2$, 0.8 M betaine and 1.5 U/µl TaKaRa RNAse inhibitor) were added to each sample, mixed gently, spun down and then incubated in a thermocycler (42℃ - 90 min, [50℃ - 2 min, 42℃ -

2 min] x 10 times, 70℃ - 15 min). For further cDNA amplification, 20 µl of PCR mix [1xKAPA HiFi Hotstart ready mix (Roche, Basel, Switzerland), 0.5 µM C1-P1-PCR-2 (5'- GAA TGA TAC GGC GAC CAC CGA T −3'), 111 nM dNTP mix] were gently pipetted to each sample, which then underwent the following thermocycler program (98℃ - 3 min, [98℃ - 20 s, 67℃ - 15 s, 72℃ - 6 min] x 30 times, 72℃ - 5 min). Amplified cDNA samples were then cleaned up of short fragments with AMPure XP beads (1:1 ratio; Beckman Coulter, Brea, CA) and checked for cDNA quality and concentration on a 2100 Bioanalyzer (Agilent Technologies, Santa Clara, CA). Samples with clear domination of long 1.5–2 kb fragments and cDNA concentrations higher than 250 pg/µl were taken for further tagmentation reaction and sequencing. Full-length cDNA libraries were tagmented and amplified with Nextera XT DNA library preparation kit (Illumina, San Diego, CA) according to manufacturer's instructions with following modifications: 75–150 pg of cDNA per sample was used as input, one tenth of all the volumes stated in the instructions were used, and the indices were diluted at a ratio of 1:3 with nuclease-free water beforehand. Aliquots of 1.5 µl of the tagmented and amplified cDNA from each sample were then pooled together as described previously (*Cadwell et al., 2016*). The sample pool was purified with AMPure XP beads according to the manufacturer's instructions, using a bead volume of 0.6x the total sample pool volume.

## Sequencing and bioinformatics

Tagmented, amplified, pooled and purified cDNA libraries with mean sizes of 550–600 bp and concentrations of 2–3 ng/µl (measured with Qubit 2.0, Invitrogen) were sequenced on a HiSeq 2500 Illumina system with single-end 50 bp read lengths. Median depth of sequencing was 4 million reads per cell. Cells, which had less than 75500 total reads, were discarded from the analysis (n = 5). Raw reads were aligned to *Mus musculus* reference genome using HISAT2 algorithm for single-end reads (*Kim et al., 2015*). Aligned reads, which unambiguously overlapped with certain gene's exons were counted using HTSeq (*Anders et al., 2015*). Both previous steps were performed in Chipster software (*Kallio et al., 2011*). Cells, where alignment performance was less than 50%, were discarded from further analysis (n = 3). To avoid amplification bias, counts were normalized between cells. This was done by finding intersecting genes that are both significantly correlated to molecular count in the reference dataset (see below) and also expressed at least one time in all PatchSeq cells. Further, expression of 150 intersecting genes found were reduced into one dimension (PC1) within the PatchSeq data and used as the normalizing factor for each cell. To avoid negative values, +one was added to the PC1 of each cell. Then gene expression was divided with this value to acquire normalized expression values for each cell.

## Clustering of the reference dataset with Seurat

As the reference dataset, we chose the mouse midbrain scRNA-seq data (*Saunders et al., 2018*) and performed Seurat clustering (*Butler et al., 2018*). 10187 cells passed quality control filter by having minimum 1000 genes and 2000 unique RNA molecules per cell. Tuning low x and y cutoff parameters to 0.4 and 0.5, correspondingly, resulted in 565 variable genes and 20 valuable principal components. Final clustering gave 19 clusters, from which only six appeared to be neuronal Sst-containing clusters.

## Mapping to the reference dataset

Bootstrap analysis with the EWCE package in R (https://github.com/NathanSkene/EWCE) was then used to map the PatchSeq dataset onto the six clusters obtained from the reference dataset. This was performed in the same way as previously described (*Muñoz-Manchado et al., 2018*), with the difference that all genes were included and the probability threshold was set as <0.05.

## Optical mapping of GABAergic local inhibitory circuits

Optical mapping was done as described previously (*Wang et al., 2007*; *Kim et al., 2014*), using a multiphoton laser-scanning imaging system (FVMPE-RS Fluoview, Olympus, Tokyo, Japan), equipped with a 25x water-immersion objective. For these experiments, Sst-ChR2-YFP male/female mice of P40-P68 age were used. No discernable changes in the light sensitivity of individual Sst neurons were detected over this age range.

We first did a series of experiments to determine the sensitivity of the ChR2-expressing Sst neurons to small spots of laser light (405 nm). For this purpose, a 510 µm by 510 µm area within the VTA of the midbrain horizontal slice was stimulated in a pseudorandom order with laser spot of different powers (5–19 µW) and durations. This area was scanned in a 32 × 32 array, giving a final pitch of 16 µm. Action potential (AP) responses were recorded from Sst neurons in current-clamp mode (n = 20), using K-gluconate-containing IS (see 'Electrophysiology' section above). By correlating the position of the laser spot with the response of the ChR2-expressing Sst neuron, it was possible to map the 'optical footprint' of the cell that defines the area of light sensitivity (*Figure 7a*; see also *Kim et al., 2014*). Photostimulation was repeated three times, and locations that evoked APs in at least 2/3 trials were included within the optical footprint of the cell. For such focal photostimulation, we used the lowest possible laser power (9 µW) that reliably evoked APs when the laser was positioned over a Sst neuron cell body, but not over its axons (*Figure 7a*). This improved the spatial resolution of the mapping and also ensured that postsynaptic responses were not caused by photostimulating axons or axon terminals, thereby allowing us to unambiguously determine the location of presynaptic ChR2-expressing Sst neurons in our circuit mapping experiments.

After optimizing photostimulating conditions, optical circuit mapping was performed by scanning the laser spot, to photostimulate presynaptic Sst neurons, while recording inhibitory postsynaptic currents (IPSCs) from a neighboring DA neuron. Inward IPSCs were recorded at a holding potential of −70 mV and input maps were constructed by correlating the location of the scanned laser spot with the amplitude of the resultant IPSCs. Input maps considered only IPSCs that were evoked in at least 2/3 trials when the laser spot was at a given location. Optical footprint and input map areas were calculated by multiplying the number of pixels by the area of a single pixel (or 256 µm$^2$). In these experiments, the IS consisted of (in mM): 140 KCl, 4 NaCl, 0.5 CaCl$_2$, 10 HEPES, 5 EGTA, 2 MgATP, 0.4 Na$_3$GTP, five disodium phosphocreatine, pH 7.2 and osmolarity 290–295 mOsm. At these conditions, calculated reversal potential for Cl$^-$ was +1.8 mV, making the IPSCs to be inward currents at −70 mV holding potential. Experiments were performed at room temperature (24°C). External aCSF solution contained 2 mM kynurenic acid and was perfused continuously. All analyses of optical mapping experiments were performed with Image J software and custom-written scripts.

## Statistics

All data are presented as means ± SEM. For electrophysiology and morphology data significance of differences between groups was analyzed using one-way ANOVA followed by posthoc Tukey's multiple comparisons test, using Prism 8.1.0 (GraphPad Software, San Diego, CA).

## Data availability

scRNA-seq raw and expression data have been deposited in the ArrayExpress database at EMBL-EBI (www.ebi.ac.uk/arrayexpress) under accession number E-MTAB-8780.

The following previously published data sets were used:

- https://www.ncbi.nlm.nih.gov/geo/query/acc.cgi?acc=GSE115746
- (*Tasic et al., 2018*)
- https://storage.googleapis.com/dropviz-downloads/static/regions/F_GRCm38.81.P60SubstantiaNigra.raw.dge.txt.gz
- (*Saunders et al., 2018*)

## Code availability

Custom written software for automated firing pattern analysis is available for downloading from here: https://github.com/zubara/fffpa (*Nagaeva, 2020a*; copy archived at https://github.com/elifesciences-publications/fffpa).

Clustering script and intermediate electrophysiological data to reproduce the results of the section 'Three electrophysiologically distinct subtypes of Sst neurons' (shown in *Table 1*, *Figure 3*, and *Figure 3—figure supplements 1–4*) can be downloaded from here: https://version.aalto.fi/gitlab/zubarei1/clustering-for-nagaeva-et.-al.-sst-vta (*Nagaeva, 2020b*; copy archived at https://github.com/elifesciences-publications/clustering-for-nagaeva-et.-al.-sst-vta).

## Acknowledgements

The following core facilities were essential for the project: Biomedicum Imaging Unit of the University of Helsinki; CSC, IT Center for Science, Finland; Genome Biology Unit, GBU, University of Helsinki, Finland; Science for Life Laboratory, SciLifeLab, Stockholm, Sweden; Canadian Neurophotonics Platform, Québec, Canada. The authors are grateful for the expert methodological and technical aid of Heidi Hytönen, Ilida Suleymanova, Martin Graf, Kelly Wong, Karen Chung and Nikolai Larin.

## Additional information

### Funding

The project was funded by the Academy of Finland (1278174 and 1317399), The Finnish National Agency for Education EDUFI, the Sigrid Juselius Foundation, and research grants MOE2015-T2-2-095 and MOE2017-T3-1-002 from the Singapore Ministry of Education. The funders had no role in study design, data collection and interpretation, or the decision to submit the work for publication.

### Author contributions

Elina Nagaeva, Conceptualization, Data curation, Formal analysis, Supervision, Validation, Investigation, Visualization, Methodology, Writing - original draft, Writing - review and editing; Ivan Zubarev, Software, Methodology, Writing - review and editing; Carolina Bengtsson Gonzales, Formal analysis, Validation, Investigation, Visualization, Methodology, Writing - review and editing; Mikko Forss, Data curation, Formal analysis, Investigation, Writing - review and editing; Kasra Nikouei, Formal analysis, Investigation, Methodology; Elena de Miguel, Conceptualization, Methodology, Writing - review and editing; Lauri Elsilä, Investigation, Writing - review and editing; Anni-Maija Linden, Investigation, Methodology, Writing - review and editing; Jens Hjerling-Leffler, Conceptualization, Resources, Data curation, Supervision, Methodology, Writing - review and editing; George J Augustine, Conceptualization, Resources, Supervision, Validation, Methodology, Writing - review and editing; Esa R Korpi, Conceptualization, Resources, Supervision, Funding acquisition, Writing - original draft, Project administration, Writing - review and editing

### Author ORCIDs

Elina Nagaeva (iD) http://orcid.org/0000-0003-2828-6234
Ivan Zubarev (iD) http://orcid.org/0000-0002-1620-8485
Lauri Elsilä (iD) http://orcid.org/0000-0002-9744-4753
George J Augustine (iD) https://orcid.org/0000-0001-7408-7485
Esa R Korpi (iD) https://orcid.org/0000-0003-0683-4009

### Ethics

Animal experimentation: Animal experiments were authorized by the National Animal Experiment Board in Finland (Eläinkoelautakunta, ELLA; Permit Number: ESAVI/1172/04.10.07/2018) and Institutional Animal Care and Use Committee in Singapore (NTU-IACUC).

### Decision letter and Author response

Decision letter https://doi.org/10.7554/eLife.59328.sa1
Author response https://doi.org/10.7554/eLife.59328.sa2

## Additional files

### Supplementary files

• Transparent reporting form

### Data availability

scRNA-seq raw and expression data have been deposited in the ArrayExpress database at EMBL-EBI (www.ebi.ac.uk/arrayexpress) under accession number E-MTAB-8780. The following previously

published data sets were used: • https://www.ncbi.nlm.nih.gov/geo/query/acc.cgi?acc=GSE115746 (Tasic et al., 2018) • https://storage.googleapis.com/dropviz-downloads/static/regions/F_GRCm38. 81.P60SubstantiaNigra.raw.dge.txt.gz (Saunders et al., 2018) • Custom written software for automated firing pattern analysis is available for downloading from here: https://github.com/zubara/fffpa (copy archived at https://github.com/elifesciences-publications/fffpa).

The following dataset was generated:

| Author(s) | Year | Dataset title | Dataset URL | Database and Identifier |
|---|---|---|---|---|
| Nagaeva E, Zubarev I, Gonzales CB, Forss M, Nikouei K, Miguel E, Elsilä L, Linden AM, Hjerling-Leffler J, Augustine GJ, Korpi ER | 2020 | PatchSeq experiment on somatostatin-expressing (Sst) neurons from mouse ventral tegmental area (VTA). Full-length single cell sequencing | https://www.ebi.ac.uk/arrayexpress/experiments/E-MTAB-8780/ | ArrayExpress, https://www.ebi.ac.uk/E-MTAB-8780 |

The following previously published datasets were used:

| Author(s) | Year | Dataset title | Dataset URL | Database and Identifier |
|---|---|---|---|---|
| Tasic B, Yao Z, Graybuck LT, Smith KA, Nguyen TN, Bertagnolli D, Goldy J, Garren E, Economo MN, Viswanathan S, Penn O, Bakken T, Menon V, Miller J, Fong O, Hirokawa KE, Lathia K, Rimorin C, Tieu M, Larsen R, Casper T, Barkan E, Kroll M, Parry S, Shapovalova NV, Hirschstein D, Pendergraft J, Sullivan HA, Kim TK, Szafer A, Dee N, Groblewski P, Wickersham I, Cetin A, Harris JA, Levi BP, Sunkin SM, Madisen L, Daigle TL, Looger L, Bernard A, Phillips J, Lein E, Hawrylycz M, Svoboda K, Jones AR, Koch C, Zeng H | 2018 | Shared and distinct transcriptomic cell types across neocortical areas | https://www.ncbi.nlm.nih.gov/geo/query/acc.cgi?acc=GSE115746 | NCBI Gene Expression Omnibus, 10.1038/s41586-018-0654-5 |

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

# Appendix 1

## Methods table

| Parameter name | Unit | Log2-transform | Used in clustering | Definition | Comment |
|---|---|---|---|---|---|
| RMP | mV | | y | Resting membrane potential (RMP) - average baseline of all sweeps (sweep - a single trace of voltage response to injected current step) in a trace. | |
| RMP std | mV | | n | | Used for outlier detection |
| Input resistance | MΩ | | y | Input resistance - regression coefficient between injected current steps and voltage responses, measured over the last 50 ms of each sweep without any action potentials (AP). | |
| SAG DP | mV | | y | SAG depolarization (DP) - ratio between steady-state mean voltage computed over the last 50 ms of the current step in sweep with 100 mV membrane voltage and maximum hyperpolarization in the same sweep. | |
| Rheobase current | pA | | y | Rheobase current - amplitude of injected current step on which the first AP appeared. | |
| Tau start | ms | | y | Tau start or Membrane time constant - base of an exponential curve fit to the −100 mV sweep. | |
| 1_spk:Spike count | n | y | y | Number of spikes. From here on **1_spk** (first spike) corresponds to the sweep evoked by rheobase current. | |
| 1_spk:First Spike Lat | ms | y | y | First Spike Latency or Delay - time interval between the beginning of the current step and that of the first AP. | |
| 1_spk:ADP amplitude | mV | | y | Afterdepolarization (ADP) amplitude - difference between the afterhyperpolarization (AHP) peak and the most positive membrane voltage during the fast repolarization phase. | Set to 0 if no ADP was detected. |
| 1_spk:ADP latency | ms | y | y | ADP latency - interval between the AHP and ADP peaks. | Set to 0 if no ADP was detected. Constant value of 0.2 (ms) was added to all estimates before log-transform. |

*Continued on next page*

*continued*

| Parameter name | Unit | Log2-transform | Used in clustering | Definition | Comment |
|---|---|---|---|---|---|
| 1_spk:AP amplitude | mV | | y | AP amplitude - difference between the AP threshold and its positive peak. | |
| 1_spk:AP HW | ms | | y | AP half-width (HW) - time interval between voltage points corresponding to half of AP amplitude during depolarization and repolarization phase. | |
| 1_spk:Thresh amplitude | mV | | y | Threshold amplitude - voltage point before the first spike after which voltage grew faster than 10 mV/ms. | |
| 1_spk:AHP amplitude | mV | | y | AHP amplitude - difference between the AP threshold and the most negative membrane potential reached during afterhyperpolarization. | |
| 1_spk:AP decay time | ms | y | y | AP decay time - Time from the AP peak to the AHP peak. | |
| Sat:Spike count | n | y | y | Number of spikes. From here on **sat** (saturation) corresponds to the sweep with the highest number of APs. | |
| Sat:Current Step | pA | | y | Saturating Current Step - injected current which evokes maximum number of APs. | |
| Sat:First Spike Lat | ms | y | y | Time interval between the beginning of the current step and that of the first AP. | |
| Sat:Fmax Init | Hz | y | y | Initial maximum frequency - Inverse of the shortest inter-spike interval (ISI) among the first three spikes. | |
| Sat:Fmax SS | Hz | y | y | Steady-state maximum frequency - Inverse of the average mean of the last three ISIs. | |
| Sat:Adaptation ratio | % | | y | Fmax SS/Fmax init | |
| Sat:AP amplitude | mV | | y | Mean difference between the AP threshold and its positive peak for all spikes. | |
| Sat:AP HW | ms | | y | Average time interval between voltage points corresponding to half of AP amplitude during depolarization and repolarization phase. | |
| Sat:Thresh amplitude | mV | | y | Mean voltage point after which voltage grew faster than 10 mV/ms for all spikes. | |
| Sat:AHP amplitude | mV | | y | Mean difference between the AP threshold and the most negative membrane potential reached during afterhyperpolarization. | |
| Sat:AP decay time | ms | y | y | Mean time from the AP peak to the AHP peak. | |

