## [Decision Letter]

**Acceptance summary:**

This manuscript describes the heterogeneity of SST+ GABAergic neurons in the mouse VTA using a combination of electrophysiology, anatomy, patch-seq, and optogenetics. The results define three basic groups of anatomically, electrophysiologically and genetically distinct neurons that are expressed unevenly through the VTA. Thus, this study provides fundamental information on VTA GABAergic cell subtypes at the molecular and cellular levels and would provide the ground for future functional studies of SST neurons. Finally, the current methodology provides an astute classification system that can be adapted to other brain structures.

**Decision letter after peer review:**

Thank you for submitting your article "Heterogeneous somatostatin-expressing neuron population in mouse ventral tegmental area" for consideration by *eLife*. Your article has been reviewed by three peer reviewers, and the evaluation has been overseen by a Reviewing Editor and Gary Westbrook as the Senior Editor. The following individuals involved in review of your submission have agreed to reveal their identity: Hong-Yuan Chu (Reviewer #1); William Wisden (Reviewer #2); John T Williams (Reviewer #3). The reviewers have discussed the reviews with one another and the Reviewing Editor has drafted this decision to help you prepare a revised submission.

Summary:

This manuscript describes the heterogeneity of SST+ GABAergic neurons in the mouse VTA using a combination of electrophysiology, anatomy, patch-seq, and optogenetics. The results define three basic groups of anatomically, electrophysiologically and genetically distinct neurons that are expressed unevenly through the VTA.

Thus, this study provides fundamental information on VTA GABAergic cell subtypes at the molecular and cellular levels and would provide the ground for future functional studies of SST neurons. Finally, the current methodology provides an astute classification system that can be adapted to other brain structures.

Essential revisions:

Please address the following points in your revised manuscript:

1) Please state in the Materials and methods whether the intrinsic excitability studies were done with the blockade of glutamatergic and GABAergic synaptic transmission?

2) How the capacitance was measured?

3) Optogenetic mapping studies, IPSCs may be due to stimulation of GABAergic axon terminals and/or cell bodies, so it not accurate to say how many SST neurons converge onto DA neurons, i.e., it can be GABA release from axon terminals of the same neuron. I would suggest the authors revise related text.

4) What is the absolute laser power (mW or mW/mm^2) at 2% or 5% of stimulation in the current study?

5) In optogenetic mapping studies, the SST-ChR2-eYFP mice used were P40-65. Whether the authors checked the expression level of ChR2 or morphology of axons in SST+ cells within this 4-week time window? Dynamic expression levels of ChR2 and/or axon atrophy associated with prolonged ChR2 expression are potential technical issues.

6) The author claimed that HFF neurons are the most excitable neurons among three SST subtypes, but only ADP can be optogenetically excited. Thus, the authors should not claim which subtype is more excitable based on somatic current injection. Particularly, somatic and dendritic excitability could be different.

7) Patch-seq data are easy to be contaminated by technical confounds (Tripathy et al., 2018). Please provide more details on quality control standards, such as if any negative controls were done to exclude confounds during RNA extraction from brain slices.

8) The quality of figures should be improved (e.g. The numbering of the figures is mixed up).

9) It is not clear that the SOM neuron subtypes would be just interneurons. Certainly, they could have only this function, but they may also send axons elsewhere. Anything known about their projections? Any projection patterns of axons outside the VTA from these SM neurons could also be used to further enhance the classification system.

---

## [Author Response]

Essential revisions:Please address the following points in your revised manuscript:1) Please state in the Materials and methods whether the intrinsic excitability studies were done with the blockade of glutamatergic and GABAergic synaptic transmission?

We were aiming to characterize somatostatin neurons in conditions closest to natural ones. Therefore, we did not use any pharmacological agents and neuronal responses were recorded in the presence of GABAergic and glutamatergic transmission. A corresponding statement has been added to the Materials and methods section (“All recordings were performed with intact GABAergic and 454 glutamatergic transmission (i.e. no pharmacological agents were added to the aCSF).”

2) How the capacitance was measured?

Cell capacitance was read from the Clampfit Software dialog window “Membrane Test” right after the whole-cell configuration was established, which is now mentioned in the Materials and methods section (subsection “Electrophysiology”, last paragraph).

3) Optogenetic mapping studies, IPSCs may be due to stimulation of GABAergic axon terminals and/or cell bodies, so it not accurate to say how many SST neurons converge onto DA neurons, i.e., it can be GABA release from axon terminals of the same neuron. I would suggest the authors revise related text.

We respectfully disagree with this point because two lines of evidence indicate that IPSCs were not caused by photostimulation of GABAergic axon terminals. First, as is typically done in such experiments, we carefully adjusted the laser intensity so that it only evoked action potentials when positioned on or near the cell body. Positioning the laser over the axon did not evoke action potentials, as is shown in Figure 7A. While it is well-known that larger and brighter light spots can evoke action potentials in ChR2-expressing axons, under our conditions this was not the case. Indeed, this is a critical prerequisite for implementing our optogenetic circuit mapping approach (e.g. Kim et al., 2014), so we were careful to design our experimental conditions to avoid this potential confound. Second, in no case has it been possible – in the hands of numerous laboratories – to use ChR2 to photostimulate neurotransmitter release directly from axon terminals in the absence of action potentials. A couple of examples from previous work done by the Augustine lab includes Wang et al., 2007, and Kim et al., 2014. Evoking transmitter release from local photostimulation of presynaptic terminals is only possible when these terminals are treated with a combination of TTX and 4-AP, to allow photostimulation to evoke presynaptic calcium spikes (e.g. Petreanu et al. (2009) Nature 457, 1142). These conditions were not used in our experiments, so we are certain that neither axonal action potential generation nor local photostimulation of axon terminals produced the IPSC responses shown in Figure 7.

In response to this comment, we have extensively revised the optogenetic mapping part of the manuscript. First, in the Results section, we have almost completely rewritten the section on photostimulation mapping of ADP-DA cell circuits (subsection “ADP cells are interneurons”). Given the reviewers’ concern about our calculation of convergence, we now avoid explicit calculation of number of converging neurons and state only that more than one ADP cell innervates a DA cell. This is a safe, conservative conclusion that is consistent with the reviewers’ comment and it also was our main conclusion in the previous version of the manuscript. Second, in the subsection on “Optical mapping of GABAergic local inhibitory circuits" in the Materials and methods section, we have more clearly spelled out how we did the mapping experiments. In particular, both in this section and in the corresponding part of the Results section we now explain (and demonstrate) that our photostimulation conditions did not result in photostimulation of axons or axon terminals. This should address the technical issue underlying the reviewers’ comment. Figure 7 and its supplements have been revised to simplify understanding of these data.

4) What is the absolute laser power (mW or mW/mm^2) at 2% or 5% of stimulation in the current study?

We have now provided the laser power values in the Materials and methods section (subsection “Optical mapping of GABAergic local inhibitory circuits”, second paragraph). Also, in the Results section (subsection “ADP cells are interneurons”, first paragraph) and in the legend to Figure 7 we spell out that the laser power was 9 μW. In the new version of the optical mapping part, we describe only results for this lower laser power (2% or 9 μW). This change slightly reduced the number of DA cells that showed evoked IPSCs (from 13 to 11), but it did not affect the main conclusion. We hope this alteration makes the results easier and clearer to follow and understand.

5) In optogenetic mapping studies, the SST-ChR2-eYFP mice used were P40-65. Whether the authors checked the expression level of ChR2 or morphology of axons in SST+ cells within this 4-week time window? Dynamic expression levels of ChR2 and/or axon atrophy associated with prolonged ChR2 expression are potential technical issues.

We agree that developmental changes in light sensitivity could potentially be a technical issue. However, we carefully compared the light sensitivity of 20 Sst neurons over this age range and did not see any clear differences in neuronal light sensitivity within this range. Therefore, we conclude that this age range would be acceptable for the mapping experiments and have now mentioned this point in the Materials and methods section (subsection “Optical mapping of GABAergic local inhibitory circuits”, first paragraph).

We did not carefully measure axonal morphology at different ages. However, no obvious abnormalities were noticed when examining ChR2-expressing neurons filled with dye. This is consistent with many previous publications using transgenic mice with neuronal ChR2 expression. For example, Arenkiel et al., 2007, stated: “Neurons expressing ChR2-YFP exhibited no sign of degeneration or altered morphology and were otherwise indistinguishable from non-expressing cells” (DOI:https://doi.org/10.1016/j.neuron.2007.03.005). Indeed, in other transgenic mouse lines expressing ChR2 in various types of neurons, we have not observed any overt neuronal or behavioral pathology even in mice that are more than 1 year old.

6) The author claimed that HFF neurons are the most excitable neurons among three SST subtypes, but only ADP can be optogenetically excited. Thus, the authors should not claim which subtype is more excitable based on somatic current injection. Particularly, somatic and dendritic excitability could be different.

We have revised the wording in the Results, when comparing the excitability properties of the neuron subtypes. (subsection “Three electrophysiologically distinct subtypes of Sst neurons”, first paragraph).

7) Patch-seq data are easy to be contaminated by technical confounds (Tripathy et al., 2018). Please provide more details on quality control standards, such as if any negative controls were done to exclude confounds during RNA extraction from brain slices.

We agree that it is hard to exclude some RNA contamination during the pipette passage through the slice. Unfortunately, we did not make any negative controls in these experiments. However, our main conclusions based on the PatchSeq data about the mixed neurochemical phenotype of Sst neurons were in agreement with the in situ hybridization results (Figure 6). Another conclusion – the DAergic nature of the Delayed neurons – was also supported by TH-positive immunohistochemical staining of these particular neurons, their morphology and electrophysiological parameters (Figures 3-6 and related figure supplements). Prevalence of the DA markers exclusively among the Delayed neurons, and not in the rest 52 neurons may indicate quality of the data. These sentences were added to the Discussion (sixth paragraph) together with the suggested reference.

8) The quality of figures should be improved (e.g. The numbering of the figures is mixed up).

We apologize for the errors in figure numbers. These errors have been corrected throughout, and figures with better resolution in.tiff format have been uploaded. Also, all figure supplements have been individually uploaded as.tiff files.

9) It is not clear that the SOM neuron subtypes would be just interneurons. Certainly, they could have only this function, but they may also send axons elsewhere. Anything known about their projections? Any projection patterns of axons outside the VTA from these SM neurons could also be used to further enhance the classification system.

The reviewers raise an interesting point about whether Sst neurons can also project, in addition to inhibiting their local targets within the VTA. We have preliminary data suggesting that at least some Sst neurons project to forebrain regions, such as the central amygdala, BNST, lateral hypothalamus and paraventricular thalamic nuclei. We have added a short note about these findings (second last paragraph of the Discussion), but opted not to present our preliminary data that will require extensive confirmation. In addition, at the moment, we do not have any data that would differentiate possible projections from the three Sst-neuron subtypes.